# Attitudes and opinions of medical practitioners, librarians, and LIS academics towards health science library services to support evidence-based medical practice in South Africa

Saroj Bala[1]*, Peter G. Underwood[2], Smangele P. Moyane[1]*

1 Department of Information and Corporate Management, Durban University of Technology, Durban, South Africa, 2 Centre for Information Literacy, University of Cape Town, Cape Town, South Africa

* sarojbalakanwal@gmail.com (SB); Smangelem1@dut.ac.za (SPM)

## Abstract

### Background

This study explores the attitudes and opinions of health science librarians, university academic staff, and medical practitioners towards health science library services to support Evidence-Based Medical Practice (EBMP) in eThekwini, South Africa. It aims to develop an EBMP model for health science library services, focusing on the importance of timely, evidence-based information for quality healthcare. The research also focuses on improving the readiness and qualifications of librarians to support medical practitioners in EBMP implementation.

### Methodology/Principal Findings

A mixed-methods survey was conducted with 251 medical practitioners (31 private and public hospitals), five health science librarians (5 public hospitals), and 24 LIS academic staff (10 South African universities and universities of technology that offer a qualification in Library and Information Science (LIS). Medical practitioners expressed tremendous appreciation for EBMP but identified considerable obstacles, such as time constraints, inadequate access to vital internet resources at their workplace, and excessive patient loads. Medical practitioners agreed that librarians can expedite their research in complex cases by providing pertinent information, yet they expressed significant dissatisfaction with current library services, identifying a deficiency in specialized EBMP support and resources. Health science librarians, though indicated, lacked formal EBMP training and were interested in taking courses or training related to EBMP, a gap reflected in LIS academic curricula, which currently offer no specialized EBMP courses.

**Data availability statement:** All relevant data are within the manuscript and its Supporting Information files.

**Funding:** The author(s) received no specific funding for this work.

**Competing interests:** The authors have declared that no competing interests exist.

## Conclusions/Significance

There is a clear gap between the documented importance of EBMP and the medical library services support available in South African hospitals. The study underscores the imperative for specialized EBMP support services training for librarians, including in LIS education and hospital professional development, alongside a significant investment in contemporary library infrastructure and digital resources. A collaborative framework for health science library services is suggested to augment the use of EBMP, better patient outcomes, and promote a culture of continuous learning.

## 1. Introduction

Evidence-Based Medicine (EBM), frequently referred to as Evidence-Based Medical Practice (EBMP), was conceptualized in the early 1990s, evolved from an emerging idea into a fundamental principle of modern healthcare delivery globally [1–3]. The shift towards EBMP, a systematic, explicit, and judicious use of current best evidence in patient care, aims to move beyond traditional knowledge-based clinical decisions [4]. This systematic approach is not only an academic exercise, but it is also important for minimizing medical errors, reducing clinical uncertainty, and confirming the delivery of ideal patient care [5]. EBMP's worldwide acceptance highlights its importance for medical professionals in directing healthcare complications and ensuring high-quality, patient-centered interventions in modern healthcare [6–8].

The digital revolution has significantly impacted health information, making EBMP a strategic priority for healthcare institutions worldwide. Health science librarians are increasingly leading medical practitioners in their search for EBMP [9,10]. The crucial and undeniable link between effective EBMP and robust, systematic literature searching clearly highlights the necessary and expanding role of librarians in this ecosystem. Health librarians have a fundamental duty to provide comprehensive, evidence-based medical information, enabling health professionals to make informed, evidence-grounded clinical decisions [11]. Librarians are vital in searching, organizing, evaluating, reviewing, and offering evidence to physicians at the moment of care [12]. Their expertise in navigating vast health information resources, joined with their expert information search and retrieval skills, positions them as essential and proactive partners in enhancing EBMP [13]. This evolving role moves far beyond traditional custodial functions, demanding critical engagement with the content and its applicability.

Rapid technological advancements, the growth of electronic information resources, and worldwide health emergencies like the COVID-19 pandemic have also led to a significant evolution in the fundamental role that health science librarians play in supporting EBMP. For the librarian to effectively support EBMP, their formal training, university curricula, and continued professional development are crucial [14]. But there hasn't been a consistent response from Library and Information Science (LIS) programs to these developments, especially in developing

countries like South Africa. Furthermore, scholarly research emphasizes the structural elements impacting health science librarians' readiness. For example, librarians must actively participate in curriculum development and assessment to support EBMP [15] and factors affecting the academic motivation of LIS students [16]. Medical librarians participate in EBMP activities pertaining to resource management and evidence dissemination, but they frequently encounter obstacles like a lack of skills, inadequate funding, and poor internet connectivity, according to research specifically from the African continent [17].

Despite the broadly acknowledged benefits and increasing adoption of EBMP, its comprehensive execution continues to face several barriers worldwide. A persistent lack of time is almost universally cited as the main barrier for medical practitioners, exceeding geographical and socio-economic boundaries [5,18–23]. Research shows librarians can significantly moderate the universal barrier to information by providing timely, synthesized, and critically appraised evidence-based information, supporting rapid clinical decision-making at the point of care [24]. Health science librarians are strategically positioned to contribute to almost every stage of the five-step EBMP process- Ask, Acquire, Appraise, Apply, and Assure- with the notable exception of the final clinical decision itself, which remains the only view of the medical practitioner [10]. Their historical evolution, which is analogous to the natural selection and adaptation of species, necessitates continuous learning, skill development, and a proactive adaptation to meet the dynamic and increasingly complex demands of modern medical practice [25]. These adaptive imperative mandates move beyond traditional library functions to develop specialized knowledge in both medical terminology and research methodologies, ultimately transforming them into indispensable and proactive information specialists [12,26].

Many recent academic studies clearly confirm the consistently favorable sentiments of medical professionals about EBMP. A Saudi Arabian study in 2022 found that despite physicians acknowledging the importance of EBMP, challenges like time constraints and limited information resources hinder its effective implementation [27]. Similarly, South Korean researchers explored that, despite strong positive opinions, a notable and significant gap remained between awareness of EBMP principles and their constant application in daily clinical practice [28,29]. This gap was often ascribed to a perceived lack of specialized skills and the availability of high-quality evidence. In light of these changing demands and technological progress, the responsibilities of health science librarians have significantly broadened. Librarians are enhancing their roles beyond information retrieval, offering systematic review services, managing research data, evaluating scholarly output's impact, and providing health professionals with advanced information literacy and critical appraisal skills [30–33]. A noticeable and increasing acknowledgment of the "embedded librarian" concept has risen, in which librarians are included directly in clinical teams. They provide essential "just-in-time" information assistance and give customized training, facilitating a more efficient knowledge transfer [34,35]. This boosted, integrated collaboration seeks to successfully close the well-known "knowledge-to-action" gap that often obstructs the successful implementation of EBMP. Moreover, medical informatics and artificial intelligence (AI) advancements offer librarians new opportunities to contribute intelligently to dataset synthesis for evidence generation and dissemination [36]. Librarians are increasingly recognized as key contributors in the creation, maintenance, and strategic distribution of institutional repositories for clinical guidelines and evidence summaries, tackling information overload [34,36].

Despite these notable achievements, health science librarians still face ongoing barriers. This includes the necessity for ongoing professional development in promptly advancing technologies, specialized medical fields, and the continual challenge of obtaining sufficient institutional funding [37,38]. The persistent digital divide in numerous developing nations, particularly in substantial regions of South Africa, remains a significant obstacle, severely affecting equitable access to online resources and the essential infrastructure required for providing advanced library services [39]. The increasing demand for librarians with expertise in bioinformatics, exact research data management, and specialized systematic review techniques has increased, placing significant pressure on LIS academic programs to rapidly update and enhance their curricula. This underscores the imperative for academic institutions to synchronize their curricula with modern professional requirements [36,38].

In the South African context, although the fundamental problems outlined in previous research continue to exist, there is a noticeable and increasing academic and professional dialogue focused on robustly enhancing the national EBMP ecosystem [40–42]. It was highlighted that the strategic significance of collaborative projects, such as the Collaboration for Evidence-Based Healthcare in Africa (CEBHA), lies in addressing urgent health concerns in Africa, using evidence-based strategies [43]. Recent studies have thoroughly examined the intricate incorporation of EBMP concepts into undergraduate medical education programs at several South African institutions, indicating a steady yet unfortunately inconsistent implementation [42,44]. Ongoing issues are firmly rooted in insufficient financial allocation for essential health libraries and the notable lack of formal acknowledgment for specialized EBMP roles for librarians in the wider public health sector [40].

The research highlights that despite increased global discussions on EBMP and the changing roles of information professionals, South Africa's context is often underrepresented in these discussions. Prior research in South Africa has addressed elements of EBMP knowledge and attitudes among dental practitioners [45,46] and academic healthcare [40]. However, a thorough investigation of medical practitioners' attitudes towards EBMP within hospital environments, along with their perceptions and requirements concerning health science library services, remains significantly lacking. Moreover, the readiness, training, and qualifications of health science librarians in facilitating EBMP in public and private hospitals, as well as the congruence of university LIS curricula with these evolving requirements, have not been systematically examined in a South African context. This study aims to address these significant gaps, offering a fundamental comprehension that may guide policy, training programs, and resource distribution to enhance EBMP implementation. The primary objective of this study is to establish an EBMP model for health science library services in South Africa, thereby enhancing healthcare quality and patient safety. This approach would integrate the requirements of medical practitioners with the competencies and possibilities of health science librarians, informed by the educational frameworks established by LIS academic institutions. The scope of this study is multifaceted, addressing several key objectives:

1. To identify the medical library services and resources presently offered in public and private hospitals within the eThekwini district that are intended to facilitate EBMP. This purpose seeks to delineate the current infrastructure and service provisions.

2. To understand medical practitioners' perceptions, actual utilization, and articulated requirements concerning library services at their respective hospitals. This objective aims to understand the user perspective and pinpoint discrepancies between supply and demand.

3. To ascertain the present function of health science librarians in hospitals within the eThekwini area, particularly concerning their support for EBMP. This entails evaluating their present functions and interaction levels with medical personnel.

4. To ascertain the current training and qualifications of health science librarians, as well as their self-assessed preparedness to facilitate EBMP. This objective evaluates the human resource capabilities for EBMP assistance.

5. To ascertain the scope and characteristics of training for health science librarians offered by universities in South Africa, specifically on EBMP. This analyzes the educational pipeline and its adaptability to professional requirements.

6. To ascertain the obstacles encountered by health science librarians in their endeavors to promote EBMP within the hospital setting. The purpose is to identify structural and practical barriers to efficient service delivery.

To achieve these aims, the study presents three principal research questions:

1. What are the attitudes and perspectives of medical practitioners on EBMP and their responses to health science library services that support them in EBMP? This inquiry centers on the principal end-users of EBMP and their engagement with support services.

2. What are the perspectives and views of health science librarians on evidence-based medical training and certification? This inquiry examines the viewpoints and preparedness of the principal service providers.

3. What are the perspectives and views of university academic personnel instructing in LIS regarding the readiness, training, and qualifications of health science librarians to facilitate EBMP at public and private hospitals within the eThekwini district of South Africa? This inquiry investigates the educational establishments that influence the development of the future workforce.

The findings are expected to make a substantial contribution to the current conversation on EBMP in South Africa. The research specifically seeks to offer definitive recommendations to the Health Professions Council of South Africa (HPCSA) about the incorporation and support of EBMP. The aim is to inform and enhance the overall delivery of health services by identifying significant gaps and possibilities. This study will reveal new pathways and professional opportunities for librarians to function as essential health science librarians, directly influencing patient care. The study aims to emphasize the significance of EBMP and the need for adequately funded, evidence-based medical libraries in public and private hospitals, seeking to stimulate increased attention and financial support from governmental entities, medical industries, and funding organizations. This study will significantly and uniquely enhance the existing literature on the essential role of health science librarians in promoting EBMP, particularly within the unique and growing healthcare context of South Africa. The proposed EBMP model (Fig 1) will function as a practical framework for facilitating these enhancements, delineating a collaborative approach among hospitals, librarians, academic institutions, and professional organizations to establish a more resilient and adaptive evidence-based healthcare system. In summary, the present study, therefore, aims to provide a comprehensive perspective on these systemic factors within the context of eThekwini, South Africa. By exploring the views of medical practitioners, librarians, and LIS academics, this research will inform a new model for health science library services that is specifically tailored to the local environment and its unique challenges. This approach acknowledges the need for LIS programs to equip future professionals with the skills necessary to navigate an ever-changing landscape.

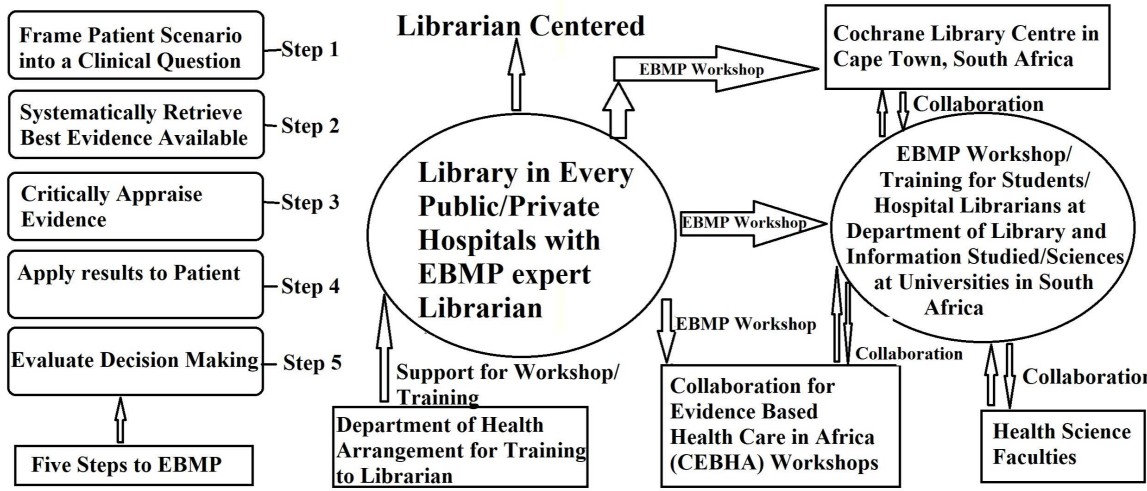

**Fig 1. Recommended EBMP model for health science services in public and private hospitals in the South African context.**

## 2. Methodology

This study conducted an in-depth investigation using a rigorous mixed-methods strategy that included both quantitative and qualitative research procedures. This intentional methodological decision sought to offer a comprehensive and detailed understanding of medical practitioners' perspectives on Evidence-Based Medical Practice (EBMP) and their complex interactions with health science library services in the eThekwini district of South Africa. The combined use of quantitative and qualitative methods enabled accurate measurement of awareness levels while concurrently investigating the underlying reasons, perceptions, and nuanced opinions that influence these patterns, thus providing a thorough and detailed understanding of the complex research issue [47].

### 2.1. Research design and approach

A survey research strategy was intentionally selected as the primary framework for this study. This design choice was based on its established effectiveness as a cost-efficient and methodical approach for collecting comprehensive data on current conditions from a representative sample of participants [48–50]. The study's quantitative component focused on the systematic collection and statistical analysis of numerical data. This facilitated the accurate evaluation of the prevalence, frequency, and distribution of specific attitudes, opinions, and professional practices associated with EBMP, primarily with closed-ended questions. In contrast, the qualitative component aimed to thoroughly investigate subjective experiences and environmental elements. The study concentrated on examining the intricacies of comprehension, the fundamental motives, and the multifaceted explanations behind observed phenomena, employing carefully designed open-ended questions to extract detailed, descriptive data from the participants. This qualitative data offered crucial contextualization and explanatory strength to the quantitative results [47–49]. The amalgamated efficacy of these methodologies offered a resilient framework to triangulate results and guarantee a more thorough and credible evaluation of the study issue. At the commencement of the study, the eThekwini district included a substantial number of healthcare institutions, namely sixteen public and twenty-three private hospitals. A vital initial study uncovered a significant discrepancy: merely six of these public hospitals have established libraries, whereas none of the private hospitals have. The survey identified nine universities and one university of technology in South Africa that provide certified Library and Information Science (LIS) qualifications for the academic component. The strategic decision to concentrate on the eThekwini district was multifaceted: it possesses a substantial and diverse hospital infrastructure, providing a comprehensive view of hospital library services and medical practice, while also enabling logistical convenience due to the researcher's geographical proximity. The intentional involvement of medical practitioners, health science librarians, and academic personnel sought to create a comprehensive and cohesive understanding of EBMP utilization, professional requirements, recognized deficiencies in current library services, and the essential element of readiness within the LIS educational framework.

### 2.2. Defining the population

The target demographic for this study was precisely defined across three separate, yet interrelated, professional groups, all located within a clearly defined geographical area, specifically the eThekwini district:

   **2.2.1. *Medical practitioners*.** This category included all general physicians and specialty medical practitioners actively involved in clinical practice in both public and private hospitals within the eThekwini area. Their participation was essential for comprehending the frontline implementation and perspective of EBMP.

   **2.2.2. *Health science librarians*.** This category includes all librarians currently employed in the designated health science libraries of public and private hospitals within the eThekwini district. Their viewpoints were crucial for comprehending the delivery of EBMP support services.

   **2.2.3. *Academic staff*.** This group included all instructional personnel engaged at higher education institutions throughout South Africa that provide professional certifications in LIS. Their observations were essential for evaluating the readiness and training of both existing and future health science librarians.

## 2.3. Sampling strategy

A complete census methodology was undertaken for the relatively limited and identifiable target populations of health science librarians and academic workers in LIS departments. This indicated that all individuals from these specified demographics were invited to participate in the study, striving for optimal inclusion and representativeness for these particular cohorts. For medical practitioners, direct random selection was impractical due to the inherent challenge of acquiring exhaustive, official listings of all practitioners from hospitals. This limitation was primarily influenced by rigorous confidentiality standards and privacy regulations. Thus, a non-probability sampling method, namely purposive/convenience sampling, was deliberately utilized for this cohort. The researcher obtained specific authorization from hospital Chief Executive Officers (CEOs) to distribute surveys during pre-arranged clinical meetings, ongoing professional education sessions, or clinical audit meetings. This strategy facilitated data collection contingent upon participant availability and willingness during sanctioned institutional gatherings, reconciling research requirements with practical accessibility limitations.

## 2.4. Data collection instruments and informal consents

The principal data-gathering tool employed among all three participant groups was a carefully designed questionnaire. The data collection started on February 1st, 2014, and ended on December 31st, 2016. The surveys were meticulously designed to address the distinct context and informational requirements of each group, e.g., medical practitioners (S1 Table in S1 File), health science librarians (S2 Table in S1 File), and academic personnel (S3 Table in S1 File). The data collection process included a two-fold strategy: personally delivered surveys were administered to medical practitioners in environments where the researcher could be present, guaranteeing elevated response rates and prompt clarification of inquiries. Simultaneously, online surveys were widely utilized, mainly for academic personnel due to their geographical distribution throughout South Africa, and for medical practitioners in hospitals, where email dissemination through a designated secretary was the sole allowable method of communication. To guarantee ethical behavior and informed involvement, thorough cover letters and extensive informational letters were distributed to all potential participants. The agreements precisely delineated the study's objective, specifically guaranteed the anonymity and confidentiality of responses, and unequivocally affirmed the voluntary character of participation, including the inherent right to withdraw at any time without prejudice. The design featured a deliberate equilibrium of open-ended and closed-ended questions to encompass a broad range of responses.

**2.4.1. *Closed questions*.** These were extensively employed for gathering demographic data and for methodically evaluating particular attitudes and viewpoints. Responses were frequently gathered utilizing a scaled response format (e.g., "Agree," "Disagree," "Neutral") or binary options ("Yes/No"). This systematic method promoted effective quantitative analysis and allowed for direct comparisons among various participant segments.

**2.4.2. *Open-ended questions*.** These were deliberately incorporated to provide participants with the essential opportunity to articulate their responses in their own terms. This qualitative aspect was essential for obtaining detailed, nuanced, and descriptive data regarding their thoughts, the particular issues they faced, and their constructive recommendations. This qualitative data was crucial in elucidating the underlying contextual reasons for the observed quantitative patterns [51].

## 2.5. Ethical approval

The ethical approval (IREC Reference Number: REC 23/13; Ethical Clearance Number: IREC 036/13), was specifically granted for the pilot study phase, which is mentioned in the manuscript in section "2.7. Pilot study, reliability, and validity," on lines 370−271, which states: "Comprehensive pilot research was undertaken before extensive data collection." The pilot study was successfully completed in July-August 2013. Following the completion of the pilot study and the refinement of data collection tools, the researcher applied for ethical approval for the final study. This subsequent approval

was granted by the Institutional Research Ethics Committee (IREC) at Durban University of Technology on September 30, 2013 (IREC Reference Number: REC 23/13), stating, "We are pleased to inform you that the questionnaires have been APPROVED; you may now proceed with data collection on the proposed project." This approval covered the entire duration of our main data collection from February 1, 2014, to December 31, 2016. Further, approvals from the Health Research and Knowledge Management Component at the KwaZulu-Natal Department of Health (reference no: HRKM-128/12) and approval from the eThekwini health district (reference no: REC 23/13) were granted.

## 2.6. Ethical considerations

During all stages of this research, significant emphasis was placed on maintaining rigorous ethical standards. Formal ethical approval was meticulously secured from the Institutional Research Ethics Committee (IREC) at Durban University of Technology before the commencement of data collection. Subsequent and equally vital approvals were diligently obtained from a series of pertinent authorities, including the KwaZulu-Natal Department of Health, the eThekwini Health District, the Chief Executive Officers of the participating hospitals, and the ethics committees of all relevant universities. This complex vetting process guaranteed compliance with all institutional and federal ethical standards. Participants were assured of their identity and the complete secrecy of their responses. Stringent safeguards were instituted to guarantee that no bodily or psychological harm was expected or imposed on any participant. Participation in the study was completely voluntary, and all participants were clearly informed of their inherent right to quit at any moment without facing any adverse consequences. All obtained data were securely maintained on password-protected computer systems, with access strictly limited to the researcher and her appointed supervisors, to protect the integrity and confidentiality of the raw data.

## 2.7. Pilot study, reliability, and validity

Comprehensive pilot research was undertaken before extensive data collection. This initial phase entailed evaluating each of the three unique questionnaires with a small, representative sample of the target market, consisting of 21 medical practitioners, 6 academic personnel, and 2 librarians. The primary objectives of the pilot study were diverse: to identify potential ambiguities or issues in question phrasing, to enhance question clarity to ensure unequivocal comprehension without external aid, and to thoroughly evaluate the overall appropriateness and feasibility of the survey instruments within the designated research context [52]. Informed by the critical comments and insights obtained from the pilot findings, many iterative modifications were implemented, including the removal of extraneous questions, the explanation of subtle language, and strategic improvements to the questionnaire layout to improve user-friendliness.

Reliability, an essential component of study quality, was methodically evaluated using Cronbach's Alpha for pertinent sections of the questionnaires. A Cronbach's Alpha score of 0.70 or higher was established as the acceptable criterion for internal consistency [53]. The study attained overall reliability values of 0.735, 0.759, and 0.905 for various sections of the medical practitioner questionnaire (S4 Table in S1 File), clearly demonstrating acceptable to commendable levels of internal consistency and dependability. Validity, concerning the accuracy of the measurement instrument in assessing its intended construction, was rigorously improved by the painstaking construction of the questionnaires, assuring direct alignment with the study's specific research aims. The expert review procedure conducted during the pilot phase substantially enhanced the content and face validity of the instruments [54].

## 2.8. Data analysis

The data collected from the questionnaires were analyzed using two software packages: Microsoft Excel for preliminary organization and descriptive statistics, and IBM-SPSS v.23 for advanced statistical analysis. Quantitative data, primarily obtained via closed-ended questionnaires, was meticulously analyzed using descriptive statistics (such as frequencies

and percentages) to encapsulate essential traits and trends. Additionally, inferential statistics, particularly the Chi-square test, were utilized to evaluate the relationship and statistical significance among various variables. A $p < 0.05$ was set as the criterion for statistical significance (S5 Table in S1 File). Qualitative data obtained from the open-ended questions were subjected to extensive thematic analysis. This entailed the methodical identification, analysis, and reporting of repeating patterns, dominant perspectives, and constructive recommendations present in the textual responses. The presentation of data was meticulously crafted for clarity and accessibility, utilizing a blend of illustrative graphs and comprehensive frequency tables to convey the study's factual conclusions effectively.

## 3. Results

### 3.1. Findings from medical practitioners

A total of 251 medical practitioners from 31 public and private hospitals, including a range of general physicians and specialists, thoroughly completed the administered questionnaires, providing a substantial dataset for analysis.

#### 3.1.1. *Demographic characteristics of medical practitioners.* The demographic profile of the respondents provided essential context for interpreting their attitudes and practices related to EBMP. The participant pool was predominantly male, with 165 individuals (65.7%) identifying as male, compared to 86 females (34.4%) (S6 Table in S1 File). This gender distribution is notable and can be compared to similar studies; for instance, a higher proportion of male participants (69%) also reported in Iran [55], while a majority of female participants (64%) were found in Dubai [56], suggesting regional variations in medical workforce demographics. The implications of this gender split are explored further in the discussion, particularly regarding potential differences in work patterns and access to resources. The age distribution indicated a mature and experienced cohort. The largest segment of participants (31.5%) fell within the 31–40 age group, closely followed by those aged 41–50 (29.5%) and 51–60 (20.3%) (S6 Table in S1 File). This broad age range, with a concentration in mid-career professionals, is generally representative of practicing medical populations and implies a group with substantial clinical experience. The cross-tabulation of gender and age group (S7 Table in S1 File) revealed that male medical practitioners continued working into the 70 + age bracket, whereas the majority of working female practitioners were below 51 years. This finding might suggest differences in career longevity or work-life balance choices between genders within the South African medical profession, potentially influencing their engagement with new practices like EBMP or their ability to attend training.

In terms of professional tenure, most participants (65.7%) reported having accumulated 11 or more years of valuable experience in medical practice (31.1% with 11–15 years, 14.7% with 16–20 years, and 19.9% with 21 + years) (S8 Table in S1 File). This high level of experience is consistent with previous findings [56], where a majority (64%) of Dubai practitioners also had over 10 years of experience. The cross-tabulation of gender and years of experience (S8 Table in S1 File) further highlighted that male medical practitioners generally possessed more experience, with 74.5% having over 11 years, compared to 76.7% of female practitioners having less than 15 years of experience. This again points to potential gendered career trajectories and can influence exposure to evolving medical practices throughout their careers.

Regarding the type of hospital, 139 participants (55.3%) were working in public hospitals, while 111 (44.2%) were from private hospitals. Only one participant (0.4%) reported holding dual appointments (S9 Table in S1 File). The cross-tabulation of gender and hospital type (S9 Table in S1 File) revealed a significant association ($\chi^2 = 14.99$, df = 2, $p = .001 < 0.05$), indicating that a majority of female practitioners (72.1%) were employed in public hospitals, while male practitioners were more prevalent in private hospitals (52.7%). This finding suggests that female practitioners might prefer or be more commonly employed in public sector roles, potentially due to differing work schedules or other systemic factors. This distinction between public and private sector employment is crucial, as it often correlates with disparities in resource availability, workload, and access to professional development opportunities, all of which impact EBMP.

The year of completion of the last degree (S10 Table in S1 File) indicated that 28.3% completed their last degree between 2006 and 2010, followed by 19.9% between 2001 and 2005. Overall, approximately 60% of participants obtained their last degree before 2006. This suggests that a significant portion of the medical practitioners may not have received formal EBMP training as part of their core academic curriculum, given that EBMP principles became more formalized and integrated into medical education post-2000. This implies that many current practitioners would have acquired EBMP knowledge through continuing professional development or self-directed learning, highlighting the importance of ongoing educational support.

Medical practitioners' specializations or job titles were diverse (S11 Table in S1 File). The largest groups were general surgeons (21.5%), family medicine practitioners (15.5%), and general physicians (12.0%). This diversity reflects the broad spectrum of medical practice within the eThekwini district. A detailed cross-tabulation data for specialization by years of experience, showing varying experience levels across different fields. For example, a high proportion of medical officers (38.1%) had 0–5 years of experience, indicating entry-level roles, which might mean they are more recently exposed to foundational EBMP concepts in their training. Conversely, gynecologists (43.8%) and orthopedic specialists (45.5%) showed a higher concentration of practitioners with over 21 years of experience, suggesting more senior roles where continued professional learning is critical (S12 Table in S1 File). Cross-tabulation of specialization by gender, highlighting gender prevalence within different medical fields (e.g., 81.5% of general surgeons were male, while 61.9% of medical officers were female) (S13 Table in S1 File). Such distributions may have implications for targeted professional development initiatives.

The workload analysis highlighted the demanding nature of their profession. More than half, i.e., 153 (61%) medical practitioners, worked over 41 hours per week (S14 Table in S1 File), with a majority (69.3%) of these being male (S15 Table in S1 File). Another significant portion, 71 (28.3%), worked 31–40 hours weekly. This contrasts with some international findings; for instance, 58% of medical practitioners work more than 30 hours a week, indicating similar high workload patterns in Iran [21]. This suggests that the issue of time scarcity for engaging in EBMP is not unique to South Africa. In terms of patient load, just over one-fifth (21.1%) of participants examined more than 40 patients per day (S16 Table in S1 File), while 27.5% saw 21–30 patients daily. shows gender distribution across patient load categories, with females being the majority (55.3%) in the 31–40 patients per day category, contrasting with the overall male majority (S17 Table in S1 File). This heavy workload, where practitioners working over 41 hours a week typically examine more than 21 patients a day, is a crucial contextual factor that directly contributes to time scarcity, a significant barrier to engaging with and applying EBMP (S18 Table in S1 File).

**3.1.2.** ***Responses of medical practitioners toward EBMP and understanding of guidelines.*** The study explored medical practitioners' attitudes and opinions on EBMP through a series of seven statements designed to capture their familiarity, perceived necessity, usefulness, and intention to use EBMP, as well as their familiarity with online search engines (Fig 2). An overwhelming majority (86.8%) of medical practitioners agreed that they are familiar with EBMP. This figure is slightly higher than in a study in India [5], where 79% of participants reported familiarity. Conversely, it contrasts sharply with studies in Iran [21,57] and Bangladesh [58], where awareness levels were notably lower. This suggests a relatively high baseline understanding of EBMP among practitioners in eThekwini. A cross-tabulation of data indicated that familiarity with EBMP was not significantly associated with age ($p = 0.16 > 0.05$), meaning familiarity extended across different age groups, including senior practitioners. This suggests that knowledge of EBMP is not confined to recent graduates but is adopted across the career spectrum (S19 Table in S1 File). However, a statistically significant association was revealed between years of experience and familiarity with EBMP ($p = 0.003 < 0.05$), with a majority of highly experienced practitioners (over 11 years) reporting familiarity (S20 Table in S1 File). This might suggest learning through continuous professional development or experience rather than solely formal education, pointing to the self-directed learning component of lifelong learning in EBM [59]. Furthermore, it was demonstrated that there was a statistically significant association between hospital type and EBMP familiarity ($p = .014 < 0.05$), with private hospital practitioners

(94.6%) showing higher familiarity than those in public hospitals (80.6%) (S21 Table in S1 File). This could be due to better access to resources, professional development opportunities, or a culture of continuous learning in private settings, emphasizing resource disparities.

Crucially, 94% of participants unequivocally agreed that the application of EBMP is both necessary and useful in their specific specialization and day-to-day practice (Fig 2). This strong consensus aligns with global findings on physical therapists in the US, where 90% considered EBP necessary [20]. It was reinforced by the fact that 84.5% of those familiar with EBMP also agreed it was necessary for their practice (*p* = <0.001) (S22 Table in S1 File). Similarly, it was illustrated that 86.1% of those familiar with EBMP also found it useful in day-to-day practice (*p* = <0.001) (S23 Table in S1 File). The strong agreement between necessity and usefulness (S24 Table in S1 File, p = <.001) confirms a deep-seated belief in EBMP's practical value and relevance to their clinical work, indicating a receptive environment for its further integration.

The majority of participants (94%) also agreed that EBMP unequivocally improves the quality of patient care (Fig 2). This finding mirrors results from studies in Sweden [60] and the UK [61,62] further solidifying the perceived benefits of EBMP. This widespread belief in its positive impact on patient outcomes provides a strong motivational factor for its adoption. It was highlighted that 83.7% of those familiar with EBMP also agreed it improves patient care (*p* = <.001), indicating a consistent perception among those who understand the practice (S25 Table in S1 File). Despite this strong positive stance, 85.3% of participants expressed a need to actively increase their use of EBMP in their daily practice (Fig 2). This self-acknowledged gap between awareness/attitude and consistent application is a critical area for intervention. It implies that while they value EBMP, practical challenges prevent them from integrating it as much as they desire. It was shown that 75.7% of those familiar with EBMP also intended to increase its use, indicating a motivation to bridge this gap (*p* = .006<0.05) (S26 Table in S1 File). This suggests that efforts to provide practical support would be well-received.

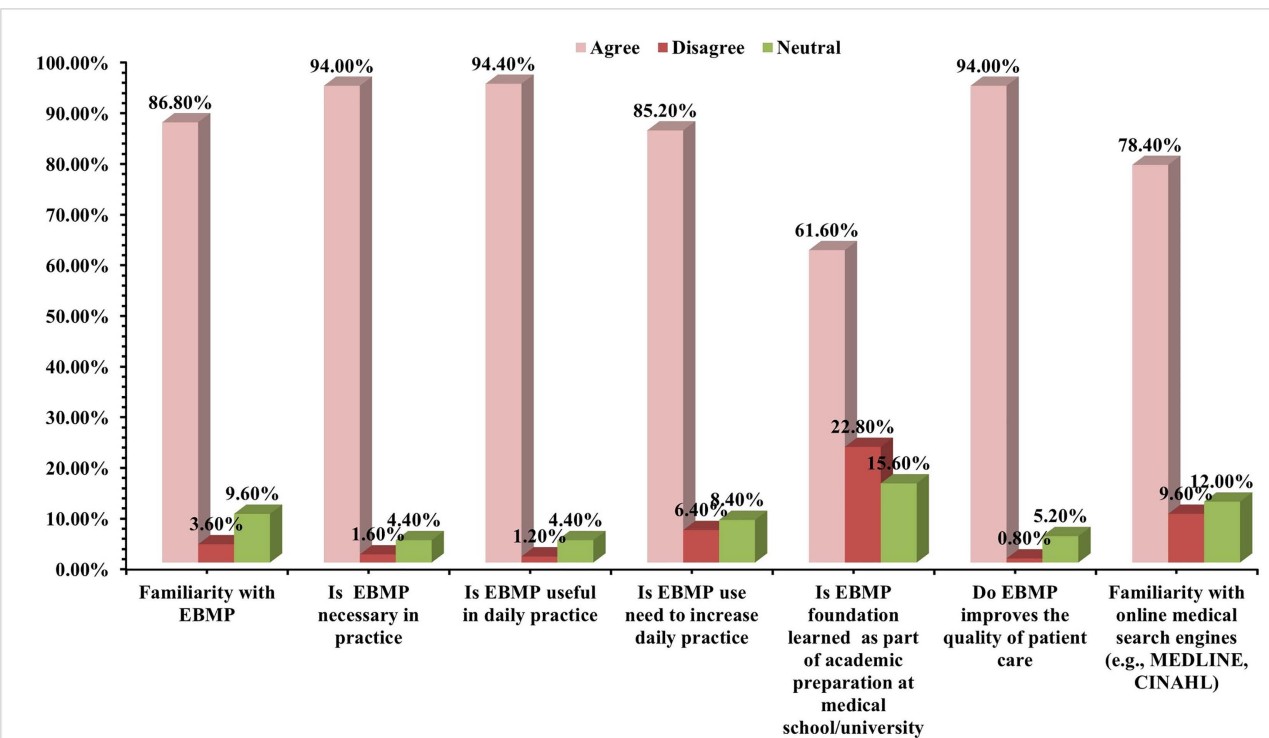

**Fig 2. Medical practitioners' responses toward EBMP and its use, benefits, and limitations.**

Regarding formal academic preparation, only 61.6% of participants reported learning the foundations for EBMP as part of their medical school/university education (Fig 2). This means nearly 40% did not, a figure comparable to a study in India (21% not exposed) [5]. This underscores a historical curriculum gap, suggesting that many current practitioners acquired EBMP knowledge through self-learning or postgraduate development rather than foundational training. This highlights the importance of continuing medical education programs and the role of information support services in bridging this initial educational deficit. Among those familiar with EBMP, only 57% learned its foundations in medical school, highlighting a significant training deficiency from their initial education ($p = .001 < 0.05$) (S27 Table in S1 File). This suggests that while EBMP is now widely accepted, its integration into core medical curricula has been a more recent development. A large proportion of practitioners (78.4%) indicated familiarity with online medical search engines (e.g., MEDLINE, CINAHL) (Fig 2). While this contrasts with lower awareness reported in some other developing countries like India, Croatia, and Bangladesh [5,58,63], it is crucial to juxtapose this theoretical familiarity with practical access. There was a significant association found between familiarity with EBMP and online search engines ($p = .001 < 0.05$), with 70.9% familiar with both. However, this familiarity does not translate to universal access (S28 Table in S1 File).

Only 107 (42.6%) out of 238 responding participants reported having internet access at their workplace to access relevant databases. A concerning 131 (52.2%) explicitly stated they did not have such access (S29 Table in S1 File). Further, a blunt difference is revealed that private hospitals provided significantly better internet facilities (60.2% access) compared to public hospitals (32.6% access), with a statistically significant association ($\chi^2 = 18.95$, df = 2, $p = <.001$) (S30 Table in S1 File). These results align with a study in Jordan (53.7% access) [64] but surpass Kuwait [65], where none had workplace internet. This digital divide within the workplace poses a significant impediment to EBMP, regardless of theoretical familiarity. Despite this, 54.1% of practitioners familiar with EBMP still lacked workplace internet access (S31 Table in S1 File), underscoring the practical challenges even for motivated individuals. This highlights a critical need for infrastructural development to support EBMP.

Regarding the necessity of EBP in medicine, almost all participants (248, or 99.6%) strongly affirmed its necessity. Their explanations consistently pointed to EBMP as "a foundation of good medical practice," essential for "maintaining current best practice and the standard of patient care," enabling "up-to-date management of medical conditions," and ensuring treatment aligns with "recent accepted guidelines and protocols." They stressed its role in keeping abreast of "current research, new developments, new approaches to treatment processes, and other latest medical achievements," as medical practice develops and changes rapidly. Without EBP, they noted, practice would regress to outdated textbooks and invalid information. These qualitative insights strongly echo the quantitative affirmation of EBMP's necessity and are consistent with previous studies [20].

When asked about sources used for EBMP, most (196, 78.1%) used both print and online/electronic sources. Only 13.1% relied solely on online sources, and 6.4% on print sources (S32 Table in S1 File). This hybrid approach indicates a transitional phase in information seeking, where traditional print media still hold relevance alongside digital resources. The results demonstrated a statistically significant association between hospital type and sources used ($\chi^2 = 18.38$, df = 6, $p = 0.005 < 0.05$), with public hospital practitioners more likely to use only print sources compared to their private sector counterparts (S33 Table in S1 File). This suggests potential resource disparities influencing practice, as public hospitals might have less access to comprehensive digital subscriptions. This mixed-media approach is consistent with a previous study [5].

Exploring deeper into print sources, it is indicated that journal readings (68%), books (53.4%), and guidelines (49.4%) were the most frequently used (Fig 3A). The use of these (4%) and atlases (5.6%) was minimal. This pattern agrees with a previous study regarding guidelines but shows lower book and journal usage compared to Indian practitioners, indicating a potential shift in preference or availability [5]. Reasons for print source preference were primarily "ease of use" (66.5%), "easily available" (41.4%), and "easy to carry" (21.5%) (Fig 3B), mirroring similar findings in India [5]. This highlights the practical advantages of print media in environments with limited digital access or comfort.

For online and electronic sources, the "free web" is most preferred (51%), followed by PubMed (48.6%), MEDLINE (42.2%), and e-journals (42.6%) (Fig 3C). The Cochrane Library was surprisingly low at 20.7%. This over-reliance on the "free web," even among those with internet access, raises concerns about the quality of evidence, as the "free web is not a suitable source of evidence unless one consults specific databases or websites of systematically reviewed articles." This pattern is similar to findings where Indian practitioners also heavily used the free web, suggesting a common challenge in identifying and prioritizing high-quality evidence in a vast digital landscape [5]. Popular websites for EBMP included Medscape (65.3%), eMedicine (27.1%), and MD Consult (28.3%) (Fig 3D).

The most significant barriers faced by medical practitioners in EBMP (Fig 4) were "lack of personal time" (85.3%) and "patient overload" (72.1%). These findings are consistent with extensive international literature [18,20,21,66]. "Lack of library services" (58.6%) emerged as the third most prominent barrier, followed by "lack of EBM training and courses" (41.8%), and "lack of resources and facilities" (25.1%). This multi-faceted challenge underscores that individual motivation is not enough; systemic support is critically missing. These barriers highlight a significant gap between the aspiration to practice EBMP and the environmental facilitators needed for its implementation.

**3.1.3. *Responses of medical practitioners toward health science library services*.** The study investigated medical practitioners' perceptions and interactions with health science library services. A critical initial finding was that the majority of participants (154, or 61.6%) reported *not* having a library in their hospital, an unambiguous contrast to the 93 (37.2%) who did. This highlights a fundamental lack of physical library infrastructure in many eThekwini hospitals, particularly private ones, as confirmed in India, suggesting a widespread deficit in basic information support systems [5]. Of those with library access (n = 93), an overwhelming majority (83, or 89.2%) expressed dissatisfaction with the library services provided. This level of dissatisfaction is substantially higher than reported in some international studies, such as

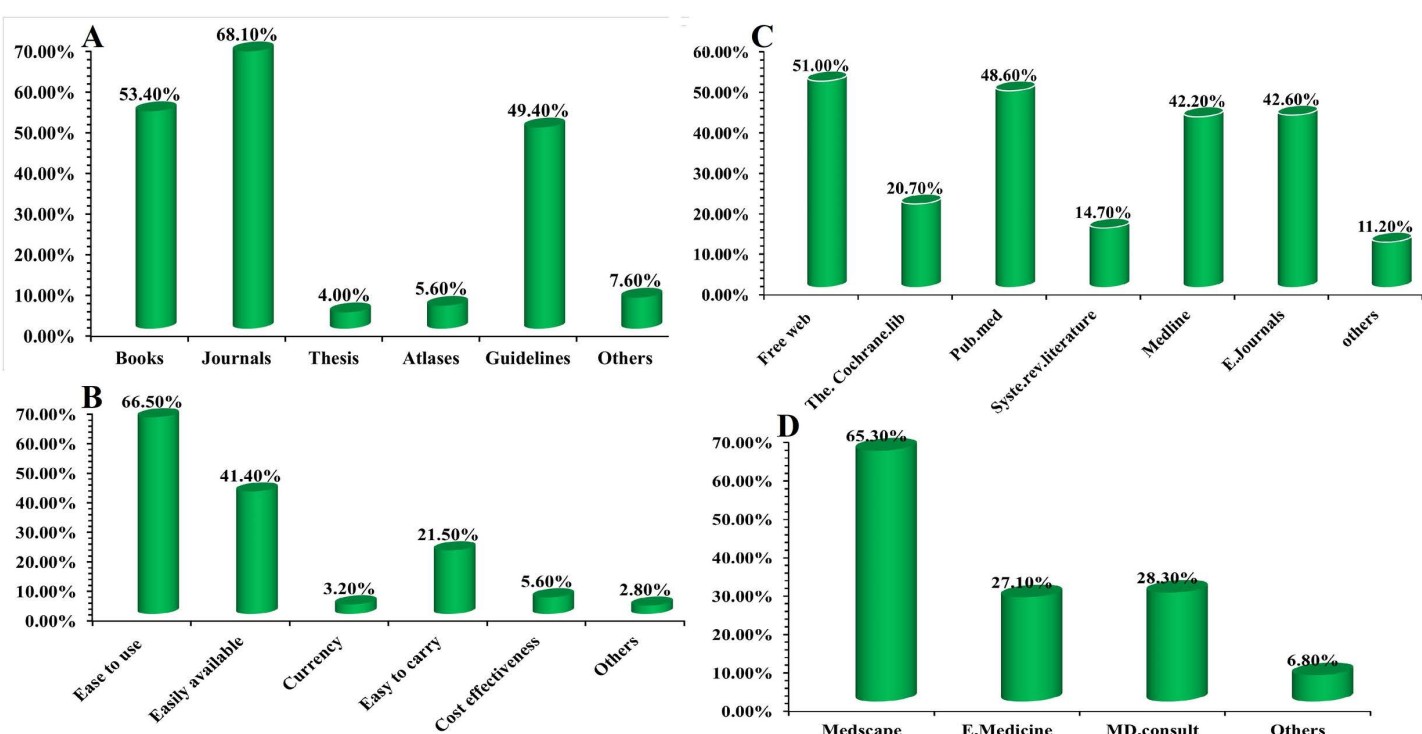

**Fig 3. (A). Types of print source used; (B), reasons for print source preferences; (C), online and electronic sub-sources; and (D), websites used to practice EBM by medical practitioners.**

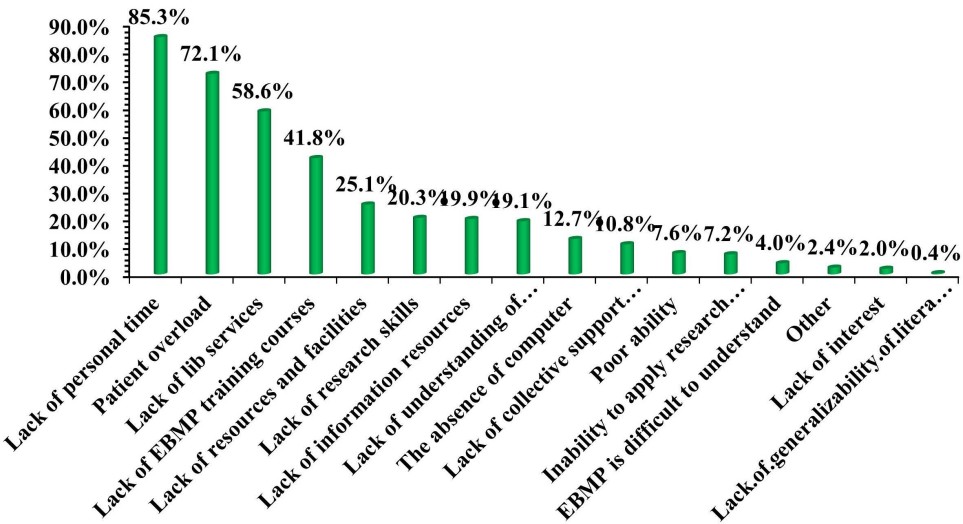

**Fig 4. Barriers faced by medical practitioners in EBMP.**

in the UK (41% satisfied) [67], indicating a severe disconnect between the current provision and user needs in eThekwini. This pronounced dissatisfaction is a strong indicator of the inadequacy of current library service models for supporting modern medical practice. When asked about desired improvements (S34 Table in S1 File), approximately half (45.7%) of the dissatisfied practitioners (n = 79) called for more books, effective computers for online searching, and, crucially, the hiring of expert librarians to assist with EBMP. Another 37.3% requested electronic resources, up-to-date journals, and access to useful medical databases. A smaller percentage desired internet access (7.2%) and extended library hours (4.8%). The cross-tabulation results (S35 and S36 Tables in S1 File) revealed that younger practitioners (20–40 age group) prioritized electronic resources, while older practitioners (above 41) more often sought expert librarians and improved book collections. This indicates evolving information-seeking preferences across generations and emphasizes the need for a multi-faceted approach to library development that caters to diverse user needs. Alarmingly, 77 (82.8%) out of 93 practitioners with library access reported that they were not assisted by librarians in their practice. Furthermore, 82 (88.2%) stated they did not have access to librarians dedicated to their specific medical field or did not perceive librarians as specialized. This perception of non-specialization and lack of assistance contrasts with findings in Iran [68], where librarians were perceived as helpful, but aligns with concerns about librarian skills in the UK [69]. This suggests a critical mismatch between what medical practitioners need and what current librarians are perceived to be providing.

When describing the actual support or services provided by librarians (S37 Table in S1 File), nearly half (47.3%) of the respondents stated "nil, not helpful," or "do not know." Only 15% reported receiving support for book searches, and 3.2% for data access. It was shown that dissatisfaction with librarian services was notably high among the 31–40 age group (32.8%), suggesting a cohort with growing EBMP needs but unmet expectations (S38 Table in S1 File). These findings starkly contrast with findings in Hamedan, Iran [68], where hospital librarians were widely considered helpful, and with US data, where librarians provided various services, including literature searches [67]. This indicates a significant gap in the proactive and specialized support offered by health science librarians in eThekwini.

The data on infrequent use of librarian services shows that only a very small percentage used services weekly (4.3%) or twice a week (4.3%). Many services were used twice a month (26.9%) or once a month (25.8%), but 14% had stopped using services due to perceived lack of usefulness (Fig 5). It was highlighted that 16.4% of those who found services unhelpful had completely stopped using the library (S39 Table in S1 File). This low utilization is in stark contrast to Iran

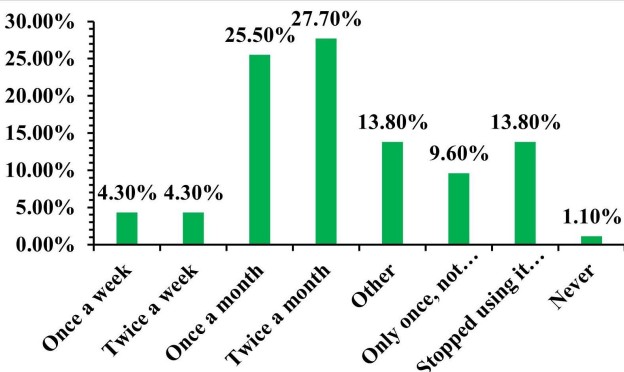

**Fig 5.  Frequency of use of librarian services by medical practitioners.**

[68] but aligns with some aspects [67] that noted non-use despite awareness. This underscores that current services are not meeting practitioners' needs, leading to disengagement. A substantial majority of medical practitioners (71%) believed that librarians were not adequately trained to assist them in their practice (S40 Table in S1 File). This perception is consistent with findings in Tehran, Iran [70], where a lack of qualified medical librarians was noted. The age group 31–40 showed particularly high dissatisfaction with librarian training (34.4% of dissatisfied practitioners in S40 Table in S1 File), which is significant given this cohort's strong interest in increasing EBMP use. Practitioners suggested that librarians should undergo training in critical appraisal, literature reviews, database searching, and EndNote, and acquire necessary EBMP qualifications (S41 Table in S1 File). These suggestions underscore the need for practical, specialized skills that go beyond traditional library science. While half of the practitioners (51.6%) believed librarians were suitably qualified for EBMP, a significant 45.2% did not (S42 Table in S1 File). This split opinion may reflect varied experiences or differing understandings of what "suitably qualified" entails for EBMP. Crucially, an overwhelming majority (92.5%) explicitly required the services of a librarian with expertise in EBMP (S43 Table in S1 File, p = 0.05). They emphasized that such expert librarians could "save their time" by assisting with specialized database searches, up-to-date research, and complex cases. This strong demand for specialized expertise aligns with calls from Tanzania [71] and Tehran [70] for expert health science librarians, confirming a universal need for this specialized role.

In hospitals without libraries, 131 (82.9%) of 158 medical practitioners strongly advocated for the establishment of library services, viewing them as a "positive asset" for housing resources, studying, and updating knowledge. They specifically requested access to key journals, free online journals, and the assistance of librarians in their research to improve patient care. This reinforces the core need for information support. Conversely, 14.6% felt no need for a library, citing workplace computers and free online sources, and expressed concerns about sustainability in private settings. This division of opinion highlights the need for both traditional library advocacy and a clear demonstration of value proposition in modern, digital contexts. For enhanced patient care without librarian services, the majority (61.4%) relied on personal computers, Google, and free online journals (S44 Table in S1 File). Other sources included university libraries (7.6%), purchasing their own books (4.4%), and discussions with colleagues (3.8%). This highlights the self-reliant, but potentially unguided, information-seeking behaviors in the absence of formal support, which may lead to variable quality of evidence accessed.

Finally, some data (S45 Table in S1 File) demonstrate a powerful "can/do" gap. While 91.6% of medical practitioners believed librarians *can* save them time, only 0.8% stated that librarians do save them time. Similarly, 90.4% believed librarians *can* assist with research in cases where little is known about a disease, but only 0.4% reported actual assistance. This consistent pattern across all statements underscores a clear latent demand for specialized librarian support

that is currently unmet. This finding strongly supports the proposition that health science librarians, with appropriate training, can significantly impact EBMP by providing timely and relevant information, as suggested previously [13,24]. Medical practitioners' "other comments" further called for trained librarians, more computer facilities, and sponsorships for global medical network access, reinforcing the quantitative findings and providing qualitative depth.

### 3.2. Findings from health science librarians

This section presents the findings from the health science librarians, a critical stakeholder group, although the small sample size (n = 5) warrants cautious interpretation and limits generalizability. All participating librarians were from public hospitals.

**3.2.1. *Demographic characteristics of health science librarians*.** The librarian cohort was predominantly female (4 out of 5, or 80%), with one male (20%). Their age range was concentrated in the 31–40 (40%) and 41–50 (60%) groups, indicating a mid-career profile. All librarians possessed substantial work experience: 60% had 6–10 years, and 40% had 11–15 years (S46 Table in S1 File). This level of experience, with 75% having over five years, aligns with findings from the US [72,73], suggesting a stable and professionally seasoned group. Despite their experience, the data on their highest qualifications showed a majority holding a National Diploma in LIS, with none updating their qualifications since 2005. This suggests a potential gap between their foundational education and the evolving demands of health information science, emphasizing the need for continuous professional development.

**3.2.2. *Responses of health science librarians toward EBMP training*.** A critical finding was that three out of five librarians (60%) explicitly stated their work does not require specialized knowledge of EBMP, which starkly contrasts with the strong demand from medical practitioners. This indicates a fundamental misalignment in job descriptions or perceived roles, suggesting that their current positions might not be formally defined to include advanced EBMP support. Furthermore, 100% of the librarians reported that they had not attended any courses or training related to EBMP or health science librarianship beyond their formal LIS qualification. This is consistent with findings in Greece [74] and Iran [12], where specialized training was lacking, highlighting a widespread educational deficit in this area.

When asked about the reasons for not attending such training, responses included lack of awareness, no opportunities, budgetary constraints, and employers' unwillingness to fund such courses. This highlights systemic barriers beyond individual motivation, suggesting that even if librarians are willing, institutional support is insufficient. Despite this, all five librarians (100%) expressed a strong willingness and desire to attend EBMP-related courses in the near future, citing their motivation to "learn more to help medical practitioners," "be more effective in their job," and "improve library services". This positive attitude towards professional development aligns with previous studies [72,74], indicating a clear readiness to adapt and contribute more effectively. In terms of their interaction with medical practitioners, four out of five librarians (80%) indicated that they do not work with specialist medical practitioners, and one reported interacting with nurses but not EBMP. All five reported providing only general services, like literature searches and medical book searches, to all hospital staff. This contrasts significantly with studies from the US, where librarians frequently provide EBM research support [72], highlighting a critical gap in direct, specialized engagement with medical practitioners in South Africa.

**3.2.3. *Responses of health science librarians toward research on EBMP resources*.** Three out of five librarians (60%) reported that their job responsibilities require expertise with EBMP resources (e.g., MEDLINE, Cochrane Library), and they possessed this expertise (S47 Table in S1 File). However, the training for these three came from varied, informal sources: the US Embassy, a previous librarian, or a supervisor. This indicates a reliance on informal, often ad hoc, training rather than structured, formalized programs, which can lead to inconsistencies in expertise and service quality. Concerning resources, three out of five librarians (60%) stated they do not have adequate resources to support EBMP, specifically citing a "lack of funds" for current subscriptions to crucial databases like MEDLINE and Cochrane Library reviews. This aligns with financial barriers reported previously [75] and emphasizes a core resource deficiency that directly impacts their ability to provide comprehensive EBMP support.

The key barriers faced by librarians in offering EBM services (Fig 6) were consistent across most respondents: "lack of organizational support" (60%), "lack of trained staff" (60%), and "lack of resources" (60%). "Lack of time" also affected 40%. These findings strongly echo previous studies highlighting that systemic and institutional issues, not merely individual capacity, impede service delivery. The qualitative comments further underscored this, with one librarian stating [72,73], "library service provision is not a priority for the Department of Health, so it is an everyday struggle to acquire resources to support EBMP," leading to "no improvement for a long time." This provides a critical explanation for the dissatisfaction expressed by medical practitioners.

### 3.3. Findings from academic staff at universities

The findings from academic staff in LIS departments, crucial for understanding the educational pipeline for future health science librarians, are presented further. Twenty-four academic staff members participated.

**3.3.1. *Demographic characteristics of academic staff*.** The gender distribution among academic staff was evenly split: 11 males and 11 females (45.8% each), with 2 non-responses (S48 Table in S1 File). The age profile indicated a concentration in mid-to-senior career stages, with the largest group aged 51–60 (37.5%), followed by 31–40 (29.2%). Most participants (45.8%) held a Ph.D. in LIS or Information Science (IS), indicating a highly qualified academic cohort. Participants represented LIS departments from various South African universities, including the University of Fort Hare (UFH), University of South Africa (UNISA), Durban University of Technology (DUT), University of Cape Town (UCT), University of Zululand (UZL), and University of KwaZulu-Natal (UKZN). This diverse representation ensures a broad perspective on LIS education in the country.

**3.3.2. *Responses of academic staff toward EBMP*.** Most academic staff (18, or 75%) were aware of EBM or EBMP, which is a positive sign for potential curriculum development. However, 6 (25%) reported no awareness, highlighting a gap even at the academic level that needs addressing for consistent educational standards. Critically, all 24 participants confirmed that their departments currently do not offer any specialized courses or training for librarians to support EBMP. This finding is consistent with reports from Kenya [76] and Tanzania [71]. The absence of specialized medical librarianship

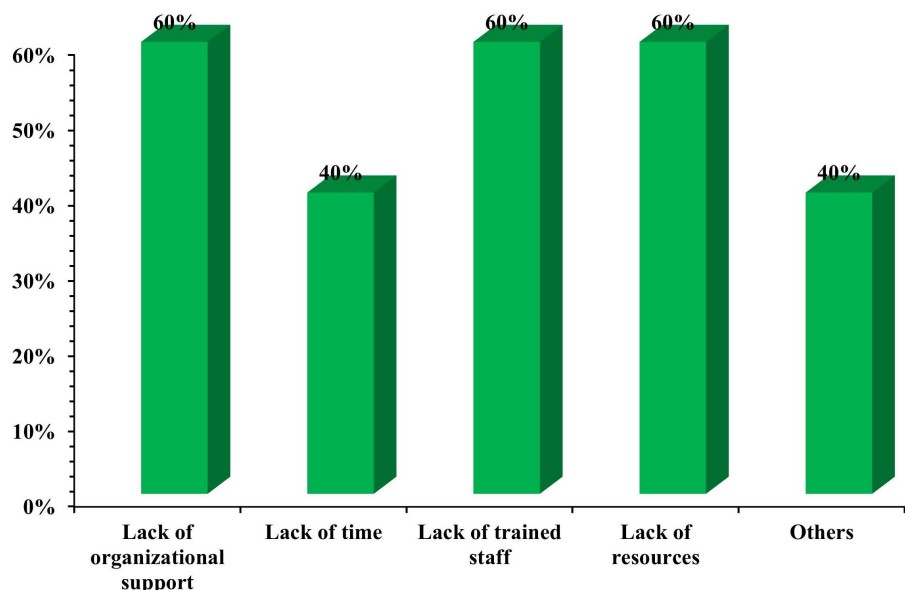

**Fig 6. Barriers faced by health science librarians in offering EBM services.**

programs was noted, indicating a regional trend. Furthermore, the majority (19, or 79.2%) indicated no plans to offer such courses in the near future, although 3 (12.5%) expressed a desire to do so. This inertia suggests systemic or resource-based challenges in curriculum adaptation, potentially due to a lack of perceived demand or resources within universities. Students rarely inquired about such courses (only 2 out of 10 responses), and medical practitioners or the health sector had never inquired. This lack of overt external demand or communication may contribute to the slow pace of curriculum change, creating a vicious cycle.

Most academic staff (21, or 87.5%) stated that students are not allowed to specialize in EBMP library services. All confirmed that only basic LIS skills are provided in undergraduate years, with no specific theoretical or practical training for EBMP. Fieldwork in medical facilities is either not required or optional. These findings collectively paint a picture of a generalist LIS curriculum that does not cater to the specific, evolving needs of health science librarianship, creating a mismatch between academic preparation and professional demands. Despite the current limitations, academic staff expressed a positive sentiment towards the *idea* of training librarians in EBMP. They believe it would be "useful in the context of South Africa regarding health practices" and suggested it could be a "post-grad qualification pursued in collaboration with the Health Science Faculty." This indicates a conceptual willingness to adapt, agreeing with a call for reviewing specialist LIS courses [76]. However, comments also hinted at potential challenges like low student enrollment and departmental capacity, underscoring the need for strategic planning and resource allocation to translate this willingness into action.

## 4. Discussion

The methodically gathered findings from this study provide the current landscape of Evidence-Based Medical Practice (EBMP) and the supporting health science library services within public and private hospitals in the eThekwini district, South Africa. Medical practitioners demonstrate an exceptionally strong and consistent positive attitude toward EBMP, clearly recognizing its necessary role in enhancing patient care and upholding current best practice standards. A universal range of significant systemic and institutional barriers unfortunately continues to hinder its optimal and widespread implementation. Alongside, the study reveals that health science librarians, despite their clear interest and demonstrated willingness to actively support this crucial area of modern healthcare, are largely unequipped with the necessary specialized training, advanced skills, and essential resources. This critical gap, furthermore, is directly continued by the current, often traditional, curricula of LIS departments in South African universities. Further, the challenges and opportunities for enhancing EBMP and health information services in South Africa's unique healthcare landscape through rigorous comparisons with existing literature are discussed.

An overwhelming familiarity with (86.8%) and reflective belief in the necessity and essential usefulness of EBMP (94%) among medical practitioners in eThekwini is an encouraging and foundational finding. This strong endorsement resonates powerfully with a consistent body of international studies that have constantly reported positive attitudes towards EBMP among healthcare professionals across diverse global contexts [23,56,77]. The undisputed agreement that EBMP directly improves patient care (94%, Fig 2) further solidifies its supposed fundamental value. This initial critical starting point clearly indicates that the primary challenge is not one of influencing practitioners of EBMP's inherent value but rather one of effectively equipping them with the practical means, necessary tools, and robust support systems required to seamlessly apply it in their demanding clinical environments. The demographic data (S6, S10 Tables in S1 File) highlights that while EBMP familiarity is present across age and experience, a significant portion of the workforce predates formal EBMP integration into curricula, underscoring the importance of post-qualification support.

Nonetheless, a notable and troubling disparity clearly arises between this exceedingly favorable disposition and the actual implementation or frequency of EBMP in routine practice. The self-reported widespread necessity to enhance EBMP utilization (85.3%, Fig 2) among practitioners clearly indicates significant underlying practical challenges. The primary barriers found in this study are "lack of personal time" (85.3%) and "patient overload" (72.1%) (Fig 4). These

findings are not isolated; they resonate profoundly and consistently with global literature, where time constraints are universally recognized as the primary and most pervasive barrier to EBMP implementation across various healthcare systems [18–21]. The empirical data indicating extensive weekly working hours (61% exceeding 41 hours, S14 Table in S1 File) and the significant daily patient load (21.1% attending over 40 patients daily, S16 Table in S1 File) within eThekwini hospitals clearly prove the widespread time scarcity and its direct effect on a practitioner's ability to engage with evidence. The statistically significant correlations among gender, age, and workload (e.g., S15, S17 Tables in S1 File) enhance our comprehension of how these systemic pressures unevenly impact various segments of the medical workforce, potentially affecting their capacity to engage with EBMP.

A significant and frequently neglected obstacle identified is the widespread "lack of library services" (58.6%) in institutional environments, particularly the prevalent "lack of information resources" (19.9%) alongside a considerable "lack of internet access" (52.2% in the workplace) (Fig 4). Although medical practitioners exhibit a theoretical understanding of major online medical search engines (78.4%, Fig 2), their practical inability to reliably access these essential resources at their workplace, especially in public hospitals (S29, S30 Tables in S1 File), results in a significant and harmful disconnect. This infrastructure shortcoming compels practitioners to commonly depend on personal resources, including private computers and unverified "free online journals" or platforms like Wikipedia for patient treatment, as indicated by 61.4% of practitioners (S44 Table in S1 File). The dependence on potentially unreliable sources generates significant concerns regarding the quality, validity, and critical evaluation of the accessed evidence, highlighting an urgent necessity for organized access to high-quality, curated resources and, importantly, thorough training in the critical appraisal of information from various sources [5]. This discovery corresponds with the persistent apprehensions over the digital gap in poor nations [39], which may restrict access to advanced research even for highly driven individuals. The observed inclination towards the "free web" (51%, Fig 3C) compared to more authoritative, systematically reviewed databases such as the Cochrane Library (20.7%) underscores a significant deficiency in information literacy and access to high-quality evidence, which directly influences clinical decision-making.

The significant dissatisfaction (89.2%) voiced by medical practitioners over current hospital library services, even among those with access, serves as a clear indictment of the existing support infrastructure. The degree of dissatisfaction is significantly greater than the levels recorded in certain overseas contexts [67], indicating a more severe issue within the eThekwini area. The principal factors contributing to this widespread discontent are distinctly recognizable: a perceived severe "absence of beneficial services" from the librarians (47.3% indicating services are "nonexistent" or "unhelpful") and a profound belief that librarians lack sufficient commitment or expertise in their respective medical domains (88.2% expressing this sentiment) (S37 Table in S1 File). This results in a disturbingly low utilization of library services, with a notably tiny fraction of practitioners indicating weekly participation (Fig 5). This clearly demonstrated this disengagement, indicating that 16.4% of practitioners who deemed services unhelpful had entirely ceased utilizing the library, underscoring the necessity for more pertinent and effective services (S39 Table in S1 File). The current perception of ineffectiveness arises from a fundamental contradiction between the conventional services provided by librarians and the evolving, evidence-based information requirements of contemporary medical practice. Although librarians meticulously execute fundamental and significant traditional functions, including basic book searching, cataloguing, and overall library management, these services unfortunately do not correspond with the intricate, time-sensitive, and specialized information needs of modern clinical decision-making. Medical practitioners clearly and consistently expressed a strong need for librarians with advanced skills to:

1. Conduct efficient research in specialist medical databases, essential due to the vast amount of information and the requirement for accuracy.

2. Perform swift and thorough literature searches for complex or uncommon situations, or for diseases with little knowledge, where prompt access to cryptic material may be critical for survival.

3. Actively support them in remaining informed about the swiftly changing research literature in their particular medical disciplines, a continual struggle due to the speed of medical progress.

4. Assume a pivotal position in delivering highly pertinent and customized information for specific patient instances, illustrating a transition from general information dissemination to individualized assistance.

5. Crucially, librarians may substantially conserve crucial time in their rigorous research activities, with 91.6% expressly affirming this (S45 Table in S1 File); time is a critical resource for busy clinicians.

The evident "latent demand" for advanced, specialist support, as previously noted [13,78], is distinctly and intensely prevalent within eThekwini's medical community. The empirical findings of this study strongly support the ongoing demand for librarians to transition from simple "source identifiers" to advanced "evidence searchers" and essential, proactive collaborators in healthcare decision-making. This requires their ability to critically evaluate, systematically arrange, and swiftly provide synthesized evidence at the point of care [11,12]. The existing service gap signifies a considerable wasted potential for augmenting patient care and optimizing clinical workflows, especially in light of the substantial workloads and patient volumes indicated by practitioners (S14 and S15 Tables in S1 File). The qualitative remarks underscore the demand for more advanced services, encompassing access to international medical networks and expert counsel.

The health science librarians openly recognize their shortcomings, with a notable 100% admitting that their existing professional qualifications have insufficiently equipped them to meet the complex requirements of EBMP. This significant shortcoming is reinforced by the absence of specialist EBMP training among the surveyed librarians, with only a scant fraction possessing pertinent fieldwork experience in medical facilities. This finding sharply contrasts with the rising aspirations and changing expectations in more developed nations, where librarians are increasingly anticipated to possess and exhibit substantial knowledge in EBMP resources and procedures [72]. It underscores a deficiency not only in individual competencies but also in the overall availability of professional growth options. The primary obstacles consistently recognized by librarians as hindering their capacity to provide full EBMP services are undoubtedly complex and deeply rooted. These factors encompass, but are not confined to, a widespread "lack of organizational support" (60%), a significant "lack of trained staff" (60%), a pronounced "lack of resources" (60%), and a continual "lack of time" (40%) (Fig 6). The striking and revealing remark from a librarian that "library service provision is not a priority for the Department of Health" expressively summarizes the systematic negligence and institutional indifference that essentially sustains these issues. Inadequate and continuous funding for specialized resources (such as essential Medline or Cochrane Library subscriptions) and focused training programs significantly restrict librarians, regardless of their enthusiasm and professional dedication, in their ability to develop and deliver necessary services. This reflects findings from various other developing nations, where funding limitations and insufficient institutional prioritization for health libraries constitute substantial and persistent obstacles [71]. The dependence on informal training highlights the absence of a systematic method for skill development in this vital domain. Despite the observed deficiencies, all participating librarians demonstrated a clear and robust eagerness to pursue specialized EBMP training, based upon the availability of opportunities, sufficient budget allocation, and appropriate training locations. This exemplified preparedness for ongoing professional development, a fundamental aspect of "lifelong learning" [79], which is a crucial and underutilized resource. It indicates that, via planned and focused investment in training and vital infrastructure, these committed librarians have the natural ability to evolve into the highly coveted "expert librarians" that medical professionals urgently require. The evident disparity between the librarians' eagerness to learn and the existing significant shortage of resources starkly underscores a crucial and ongoing impediment within the wider EBMP ecosystem in South Africa [40–42]. The library cohort's small sample size (n = 5) could lead to selection bias because the viewpoints and experiences of librarians employed by public hospitals might not be representative of those in private healthcare or other specialized library settings. Additionally, a small cohort might restrict the range of answers, possibly overrepresenting some points of view and underrepresenting others. The discussion of opportunities and challenges, for example, may be significantly biased toward the realities of underfunded public institutions, which may not be

the case in private hospitals with substantial funding. Therefore, in order to provide a more thorough and generalizable understanding of the topic, future research should strive to include a more diverse group of librarians from both public and private institutions.

The insights obtained from academic personnel at LIS departments reveal a complex and somewhat contradictory scenario. A significant majority (75%) of academic staff exhibited awareness of EBMP, although a troubling 25% acknowledged a lack of prior awareness, suggesting a potential foundational knowledge deficit even at the academic level within the discipline. This indicates a necessity for internal professional development within LIS faculties to maintain a uniform and up-to-date comprehension of advancing information science fields. The study found that none of the surveyed LIS departments provide specialized courses or training programs aimed at equipping librarians for EBMP, with a significant majority (79.2%) reporting no plans to implement such offerings in the near future. This widespread deficiency results in a substantial and adverse "curriculum gap," a condition previously observed in Kenya [76] and Tanzania [71], which observed the lack of specialist medical librarianship programs, highlighting a regional trend. Moreover, the absence of explicit demand from both LIS students and medical practitioners/health sector representatives for these specialized courses exacerbates the stagnation in curriculum development. This indicates a possible communication failure or a deficiency in understanding the capacity of LIS programs to address significant skills deficits in the healthcare industry. A majority of academic staff (87.5%) affirmed that students are prohibited from specializing in EBMP library services, and training is predominantly generalist, with fieldwork in medical institutions being optional or absent. The findings together illustrate a generalist LIS curriculum that fails to address the specialized and developing requirements of health science librarianship, resulting in a disparity between academic training and professional expectations. A three- to four-year undergraduate degree or a postgraduate diploma are the two main models that LIS programs in South Africa adhere to within the framework of LIS education. The degree to which specialized subjects like EBMP are covered can be influenced by this structure, especially given how generalist many undergraduate programs are. Research methods, information literacy, and knowledge management are among the fundamental skills that are well-established in LIS curricula. Nonetheless, there seems to be a sizable lack of specialized education that particularly addresses the particular requirements of EBMP and health librarianship. The curriculum may not sufficiently address the application of these skills in clinical settings, such as using particular medical databases (e.g., PubMed, CINAHL) or navigating the complexities of institutional ethics boards and health data privacy regulations (e.g., HIPAA), even though students are taught how to conduct literature reviews and synthesize information. The results of this study are thus in line with more general discussions in the literature on LIS education, which have long argued over how to strike a balance between a generalist and a specialist curriculum. According to studies, LIS programs may still not offer the in-depth instruction needed for specialized fields like health information science, even though they have changed to incorporate more technology. Competencies in areas such as research data management, data analytics, and educational technologies, which are not always central to traditional LIS training, are necessary for the changing role of the health librarian, especially in light of the growth of digital health after 2016.

Notwithstanding these evident current constraints and the lack of official curricula, the favorable remarks expressed by the academic staff concerning the concept of training librarians in EBMP are definitely encouraging. They assert it would be "beneficial in the context of South Africa concerning health practices" and propose it may be a "postgraduate qualification" undertaken "in partnership with the Health Science Faculty." This signifies an acknowledgment of the changing professional environment and the necessity for transformation, despite the fact that the practical execution of these curricular reforms is still in its early phases. Recent academic research strongly emphasizes the pressing need for curricular reform [40–42]. Some researchers underline the necessity for South African LIS schools to promptly incorporate advanced competencies, including bioinformatics and complex systematic review methodologies, into their curricula to effectively address the swiftly changing requirements of the health information sector [36,38,40–42]. Global trends indicate that LIS programs are proactively adapting to include new roles, such as data management librarians and embedded information, which may serve as viable models for implementation within the South African academic context [30,80]. The

current issue is to convert this intellectual acknowledgment into concrete and meaningful curricular improvement, maybe propelled by enhanced advocacy and participation from the healthcare sector.

Simultaneously, it is important to discuss the differences between private and public sector settings, explaining disparities in resource access and practitioner attitudes. There is an unambiguous difference between the private and public sector healthcare systems in South Africa. The public sector, funded by tax revenue, serves the majority of the population (≈84%), while the private sector, funded by medical aid schemes, serves a wealthy minority (≈16%). Due to the substantial concentration of resources, such as sophisticated technology, infrastructure, and human capital, in the private sector as a result of this funding gap, the public system is consistently overburdened and underfunded. Practitioner attitudes regarding EBMP implementation may be impacted by this disparity in resources. Government healthcare workers frequently deal with serious issues like a lack of employees and obsolete equipment, which can result in high levels of stress at work, burnout, and professional frustration that could be interpreted as disengagement or a negative viewpoint [81,82]. Conversely, practitioners in the private sector enjoy the advantages of a well-resourced setting that enables them to concentrate more on patient care and professional growth.

Private sector practitioners also benefited from having seamless access to the latest research databases, journals, and digital tools as compared to public sector practitioners who often struggle with unreliable internet connectivity and lack of access to databases, which hinders their ability to engage with new research and apply it to clinical practice [83]. Hence, these disparities in resource access and practitioner attitudes are a direct consequence of this systemic inequality. It requires fundamental changes in the health system to ensure equal distribution of resources. The proposed National Health Insurance (NHI) aims to merge the system and bridge this gap, but its execution remains a significant challenge [84]. By acknowledging these complex systemic factors, this study provides a more nuanced understanding of the challenges faced by health librarians and practitioners, highlighting the need for targeted interventions that go beyond individual professional development and address the root causes of inequality.

Lastly, it is important to discuss whether the data collection period (2014–2016) may represent a limitation, particularly when there are major developments in digital health and library services, particularly after COVID-19. Since 2016, South Africa has developed a National Digital Health Strategy (2019–2024) aimed at creating new technology to improve healthcare services [85]. The period forced many academic and health science libraries to familiarize themselves with virtual services, enhance online reference services, and integrate their aids with learning management systems. Therefore, the findings of this study, while providing a valuable baseline of attitude and perceptions from the pre-pandemic era, may not fully reflect the current landscape. Thus, this study should be viewed as a foundational analysis highlighting historical challenges and opportunities that may have been aggravated or transformed by more recent events. The insights gathered, however, remain relevant for understanding the systemic factors that continue to influence EBMP support in South Africa and can inform future research into the post-pandemic state of health science libraries.

## 5. Conclusion and implications

This study definitively demonstrates that medical practitioners in the eThekwini district of South Africa exhibit a high degree of awareness and a markedly positive attitude towards EBMP, acknowledging its essential role in improving patient care. Nonetheless, their capacity to comprehensively execute EBMP is profoundly obstructed by substantial structural obstructions, including time limitations resulting from excessive workloads and, most crucially, a widespread deficiency in health science library services and access to vital information resources. Present library services are predominantly fundamental and inadequately address the specific requirements of practitioners, resulting in significant dissatisfaction and underutilization. Health science librarians, despite their eagerness and commitment, are presently inadequately prepared to deliver essential EBMP support, particularly due to a deficiency in specialized training and limited institutional resources. This shortfall is exacerbated by LIS academic programs that, although theoretically receptive to EBMP training, currently lack specialized curricula to equip future health science librarians. The findings underscore a significant disparity

between the demand for and the provision of appropriate EBMP support in South African hospitals. A multifaceted and collaborative strategy is required to overcome this gap and cultivate an evidence-based healthcare system.

## 6. Recommendations for future development and research

Based on the robust empirical findings and comprehensive conclusions drawn from this study, the following actionable and strategic recommendations are put forth to foster a more resilient, responsive, and effective EBMP environment within South African public and private hospitals, with particular emphasis on the eThekwini district and broader national implications. These recommendations also outline critical areas for future research to address the limitations identified:

### 6.1. Prioritized establishment and resourcing of fully functional, EBMP-focused hospital libraries

It is recommended that comprehensive, modern library services, staffed with dedicated and specialized librarians, be systematically established in every public and private hospital across the nation. These libraries must be adequately equipped with up-to-date resources and robust technological infrastructure. This can be justified, as this study unequivocally demonstrated a profound and unmet need from medical practitioners, particularly within private hospitals, for such facilities. They recognize the pivotal role these libraries can play in facilitating knowledge acquisition, research access, and ultimately, improving patient care. This recommendation is in strong alignment with long-standing international calls for every hospital to possess a well-equipped library staffed by evidence-based medical librarians. Sustained funding, encompassing subscriptions to critical databases, access to current journals, and reliable internet connectivity, is non-negotiable.

### 6.2. Mandatory and comprehensive specialized EBMP training for health science librarians

Rigorous, comprehensive, and ongoing training programs specifically focused on EBMP must be mandated and provided to all health science librarians. The Department of Health and private hospital groups bear the responsibility for allocating dedicated budgets and proactively arranging for the delivery of such essential professional development. This is justified from this study, as a critical finding was the unanimous desire among surveyed librarians for such specialized training, juxtaposed against their current lack of opportunity and funding. This training is fundamental to equipping them with advanced skills in critical appraisal, systematic searching of complex medical databases (e.g., MEDLINE, Cochrane Library), efficient information synthesis, and proficient use of specialized software like EndNote. Addressing this training deficit directly resolves the current negative perception of library services and empowers librarians to transition into proactive, indispensable information partners.

### 6.3. Proactive curriculum reform and inter-faculty collaboration in LIS education

LIS departments in South African universities must urgently and proactively review, update, and reform their curricula to include specialized modules, concentrations, or dedicated streams in health science librarianship, with an explicit and robust emphasis on EBMP. This necessitates sustained and synergistic collaboration with health science faculties and established expert bodies like the Cochrane Library Centre in South Africa. This study justifies it as academic staff acknowledge the vital importance of EBMP training for librarians, yet current LIS curricula consistently fail to adequately prepare students for this specialized role. Such inter-faculty collaboration will ensure clinical relevance and practical applicability and will equip future health science librarians with the necessary competencies to make a tangible and effective contribution to EBMP. Making practical fieldwork in diverse medical libraries and healthcare facilities a mandatory component of LIS education is also crucial for experiential learning.

### 6.4. Formal recognition and strategic integration of EBMP librarians within healthcare teams

The Health Professions Council of South Africa (HPCSA) and other pertinent healthcare regulatory and professional authorities must formally recognize, legitimize, and endorse the critical and specialized role of health science librarians

in actively supporting EBMP. This formal recognition should pave the way for their deeper integration into healthcare teams, ideally through the widespread adoption of an "embedded librarian" model. This is justified as medical practitioners overwhelmingly believe librarians can significantly save them time and provide invaluable assistance in navigating complex clinical situations. Formalizing this role would facilitate better resource allocation, establish clear career pathways for specialized librarians, and foster closer, more seamless collaboration between medical and information professionals, enhancing the efficiency and effectiveness of patient care.

### 6.5. National prioritization of health information infrastructure and digital resources

Beyond the establishment of physical libraries, there must be a substantial and sustained national investment from both the Department of Health and private hospital management in providing ubiquitous, reliable, and high-speed internet access across all hospital premises. This must be complemented by centrally funded subscriptions to essential, high-quality medical databases (e.g., MEDLINE, Cochrane Library, UpToDate) that are readily accessible from every medical practitioner's workstation. It is justified from this study, as the lack of consistent workplace internet access and adequate access to relevant databases emerged as a significant barrier to EBMP. This investment in foundational digital infrastructure is not a luxury but a fundamental prerequisite for enabling practitioners to efficiently access and utilize evidence effectively, even before direct librarian intervention. Without reliable access to the evidence itself, the most skilled librarian cannot fully empower EBMP.

## 7. Future research directions

Building upon the valuable insights collected from this study and critically addressing its acknowledged limitations, the following areas are recommended for future research to further advance the understanding and implementation of EBMP and the role of health science librarians in South Africa:

### 7.1. Longitudinal studies on EBMP adoption and library engagement

Conduct extended longitudinal studies to systematically track changes in medical practitioners' EBMP practices, their attitudes, and their sustained engagement with health science library services over an extended period. Such studies would be particularly valuable if initiated following the implementation of any of the recommended interventions (e.g., specialized librarian training, new library establishments). This would provide crucial data on the long-term impact and sustainability of proposed changes.

### 7.2. Exploring medical and library student perspectives on EBMP

Initiate comprehensive survey studies specifically among medical students to ascertain their interest in the formal integration of EBMP into their undergraduate and postgraduate curricula, and to understand their expectations regarding future librarian support in their clinical practice. Concurrently, conduct similar surveys among current library and information science students and nascent researchers to gauge their interest in pursuing specialized careers as health science librarians and to understand their perceptions of the requisite specialized training and competency frameworks. This would inform the recruitment and foundational training pipeline.

### 7.3. Rigorous impact assessment of proposed interventions

Design and implement rigorous interventional studies to evaluate the concrete effectiveness of proposed strategies. This includes assessing the impact of specialized EBMP training programs for librarians, the establishment of new hospital library services, or the implementation of embedded librarian models on measurable outcomes such as actual EBMP uptake by practitioners, improvements in clinical decision-making, and, ultimately, enhanced patient outcomes. Quantitative metrics and qualitative assessments would be essential.

### 7.4. In-depth qualitative exploration of barriers and facilitators

Complement quantitative findings with more in-depth qualitative research. Conduct focused ethnographic studies, detailed case studies, or extensive focus group discussions and semi-structured interviews with a larger and more diverse sample of medical practitioners and health science librarians (including those in currently underserved private hospitals and rural areas). This would allow for a richer, more nuanced understanding of the deeply embedded psychological, social, organizational, and cultural barriers, as well as the key facilitating factors, to EBMP implementation within specific South African contexts.

### 7.5. Comparative studies across diverse South African regions

Expand the geographical scope of future research beyond the eThekwini district to encompass other provinces and diverse rural areas within South Africa. This would facilitate comparative analyses to identify regional disparities in EBMP knowledge, prevailing attitudes, resource availability, and the nature of existing library support. Such comparative data are crucial for informing and developing more tailored, equitable, and effective national strategies for EBMP dissemination and support.

### 7.6. Economic cost-benefit analysis of EBMP support

Conduct rigorous economic studies to quantify the tangible benefits and return on investment associated with increased investment in specialized health science library services and comprehensive EBMP training for medical practitioners. This analysis should aim to quantify outcomes such as reduced medical errors, improved patient safety, enhanced clinical efficiency, better health outcomes, and the monetary value of time saved for highly skilled medical professionals. Providing a strong, evidence-based economic case is critical for advocating increased funding and resource allocation from policymakers and healthcare administrators.

### 7.7. Limited number of library cohort

It is acknowledged that the small librarian cohort (n = 5), all from public hospitals, severely limits the representativeness and generalizability of librarian-related findings. Therefore, for future studies, more librarians from public and private hospitals must be identified and included.

## Supporting information

**S1 File. Table S1. Questionnaire for medical practitioners. Table S2. Questionnaire for health science librarians. Table S3. Questionnaire for academic staff at the university. Table S4. Cronbach's alpha results. Table S5. Chi-square test results. Table S6. Demographic characteristics of medical practitioners. Table S7. Gender * Age Cross tabulation. Table S8. Gender vs Years of experience Cross tabulation. Table S9. Gender * Type of hospital Cross tabulation. ($\chi$2 = 14.99, df = 2, *p*-value = .001 < 0.05). Table S10. Year of completion of medical practitioners' last degree. Table S11. Medical practitioners' specialisation or job title. Table S12. Medical practitioners' specialisation or job title * Years of experience Cross tabulation. Table S13. Medical practitioners' specialization or job title * Gender Cross tabulation. Table S14. Average hours of weekly practice. Table S15. Average hours of weekly practice * Gender Cross tabulation. Table S16. Average number of patients daily examined. Table S17. Average number of patients daily examined * Gender Cross tabulation. Table S18. Average hours of weekly practice * Average number of patients daily examined Cross tabulation. Table S19. Age * Familiarity with EBMP Cross tabulation. ($\chi$2 = 14.16, df = 10, *p*-value = 0.16 > 0.05). Table S20. Years experience in medical practice * Familiarity with EBMP Cross tabulation. Table S21. Type of hospital where medical practitioners work * Familiarity with EBMP Cross tabulation. ($\chi^2$ = 12.43, df = 4, *P*-value = .014 < 0.05). Table S22.**

Familiarity with EBMP * Recognition that EBMP is necessary for specialisation Cross tabulation. ($\chi 2$ = 26.77, df = 2, *P*-value = <0.001). Table S23. Familiarity with EBMP * Recognition that EBMP is useful in day-to-day practice Cross tabulation. ($\chi 2$ = 78.36, df = 4, *P*-value = < 0.001). Table S24. Recognition that EBMP is necessary for specialisation * Recognition that EBMP is useful in day-to-day practice Cross tabulation.($\chi 2$ = 175, df = 4, *P*-value = < .001). Table S25. Familiarity with EBMP * EBMP improves quality of patient care Cross tabulation. ($\chi 2$ = 23.80, df = 4, *P*-value =<.001). Table S26. Familiarity with EBMP * Intention to increase EBMP in daily practice Cross tabulation. ($\chi 2$ = 14.26, df = 4, *P*-value = .006 < 0.05). Table S27. Familiarity with EBMP * Learned foundations for EBMP at medical school/university Cross tabulation. ($\chi 2$ = 19.76, df = 4, *P*-value = .001 < 0.05). Table S28. Familiarity with EBMP * Familiarity with online medical search engines (e.g., MEDLINE, CINAHL) Cross tabulation. ($\chi 2$ = 17.80, df = 4, *P*-value = .001 < 0.05). Table S29. Access of relevant databases and the Internet at work place. Table S30. Hospital where participants are currently working * Access of relevant databases and the Internet at workplace Cross tabulation. Table S31. Familiarity with EBMP * Access to relevant databases and the Internet at workplace cross tabulation. Table S32. Sources used by medical practitioners to practice EBM. Table S33. Type of hospital where medical practitioners work * Sources used by medical practitioners to practice EBM. Cross tabulation. Table S34. Medical practitioners' suggested improvements to library services. n = 79/83. Table S35. Age group * Medical practitioners' suggested improvements to library services Cross tabulation. Table S36. Age group * Medical practitioners' suggested improvements to library services Cross tabulation. Table S37. Support or services provided by librarians. n = 67/93. Table S38. Age group * Support or services provided by librarians Cross tabulation. Table S39. Support or services provided by librarians * Frequency of use of librarian services Cross tabulation. Table S40. Age * Medical practitioners' perception of training of librarians to assist with EBMP Cross tabulation. Table S41. Medical practitioner' opinions on training of librarians to assist with EBMP. Table S42. Familiarity with EBMP * Medical practitioners' perception of training of librarians to assist with EBMP Cross tabulation. Table S43. Familiarity with EBMP * Medical practitioners' requirement of a librarian with expertise in EBMP Cross tabulation. Table S44. Medical practitioners' other information sources for patient care. Table S45. Statements of medical practitioners about librarians. Table S46. Gender, age, experience, and job title of health science librarians. Table S47. Health science librarians' expertise with EBMP resources. Table S48. Gender and age of academic staff.
(PDF)

## Acknowledgments

Durban University of Technology for the Ph.D. scholarship to Saroj Bala.

## Author contributions

**Conceptualization:** Saroj Bala.

**Data curation:** Saroj Bala.

**Formal analysis:** Saroj Bala.

**Investigation:** Saroj Bala.

**Methodology:** Saroj Bala.

**Software:** Saroj Bala.

**Supervision:** Peter G. Underwood.

**Validation:** Saroj Bala.

**Visualization:** Saroj Bala.

**Writing – original draft:** Saroj Bala.

**Writing – review & editing:** Saroj Bala, Peter G. Underwood, Smangele P. Moyane.

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
