## [Decision Letter · Decision Letter 0]

6 Aug 2025

Dear Dr. Bala,

Thank you for submitting your manuscript to PLOS ONE. After careful consideration, we feel that it has merit but does not fully meet PLOS ONE’s publication criteria as it currently stands. Therefore, we invite you to submit a revised version of the manuscript that addresses the points raised during the review process.

We look forward to receiving your revised manuscript.

Kind regards,

Farshid Danesh, Ph.D.

Academic Editor

PLOS ONE

Journal Requirements:

2. Please remove all personal information, ensure that the data shared are in accordance with participant consent, and re-upload a fully anonymized data set.

Reviewers' comments:

Reviewer's Responses to Questions

**Comments to the Author**

1. Is the manuscript technically sound, and do the data support the conclusions?

Reviewer #1: Yes

Reviewer #2: Partly

Reviewer #3: Yes

2. Has the statistical analysis been performed appropriately and rigorously?

Reviewer #1: Yes

Reviewer #2: Yes

Reviewer #3: Yes

3. Have the authors made all data underlying the findings in their manuscript fully available?

Reviewer #1: Yes

Reviewer #2: Yes

Reviewer #3: Yes

4. Is the manuscript presented in an intelligible fashion and written in standard English?

Reviewer #1: Yes

Reviewer #2: No

Reviewer #3: Yes

Reviewer #1: Hello

It's a good article and I don't think there's anything wrong with it, but I have two comments: the title of the article should be a little shorter, and secondly, the keywords for working with mesh should be standardized.

Reviewer #2: This manuscript presents a comprehensive mixed-methods study exploring the support structures for Evidence-Based Medical Practice (EBMP) within health science libraries in the eThekwini district, South Africa. It addresses important gaps regarding librarian training, library resources, and practitioners’ engagement with EBMP. The topic is highly relevant to global health information science and library services, and the manuscript contributes valuable localized data with implications for both policy and LIS curriculum reform.

1. The Introduction would be strengthened by stepping further back to discuss the foundational roles of student education, university curricula, and formal training in preparing health science librarians for Evidence-Based Medical Practice (EBMP) support. Adding context about how LIS programs have (or have not) adapted to rapid technological changes—including the impacts of global pandemics and advances in electronic information resources—would help readers better understand the environment shaping current challenges. Highlighting these historical and educational dynamics early will ensure that all readers have a comprehensive perspective on the systemic factors influencing EBMP support in South Africa. Integrating recent and relevant international literature is recommended to solidify this context. It is advisable to reference relevant literature such as: 10.5334/pme.1145, https://digitalcommons.unl.edu/libphilprac/5340, 10.1080/02763869.2015.1082375, 10.1080/02763869.2024.2370756

2. The librarian cohort (n=5) is very small, all from public hospitals, severely limiting the representativeness and generalizability of librarian-related findings. This should be acknowledged explicitly as a study limitation and, if possible, discussed in terms of how it might bias results.

3. The data collection period (2014–2016) is significantly outdated considering the rapid evolution of digital health resources and library services, particularly post-COVID-19. The manuscript should better address the potential impact of this time lag on current relevance.

4. While academic staff perspectives are included, the analysis of LIS curricula is somewhat superficial. Consider including more detailed curriculum reviews or syllabi analysis to substantiate claims regarding educational gaps.

5. The manuscript discusses differences between private and public sector settings but sometimes simplifies complex systemic factors. A more nuanced discussion could better explain disparities in resource access and practitioner attitudes.

6. Some sentences are overlong or complex, affecting readability. The manuscript would benefit from professional copyediting to improve clarity and conciseness.

Reviewer #3: The present article is challenging only in terms of the statement of the problem and the title.

My suggestion is to make the title shorter. This title seems too long. Secondly, I suggest that the research gap be presented in a very precise and clear manner. Overall, the rest of the article is technically and scientifically sound in my opinion.

**Do you want your identity to be public for this peer review?** For information about this choice, including consent withdrawal, please see our Privacy Policy

Reviewer #1: No

Reviewer #2: No

Reviewer #3: No

---

## [Author Response · Author response to Decision Letter 1]

14 Aug 2025

Date: 15 August 2025,

Subject: Manuscript (PONE-D-25-34561) – Attitudes and Opinions of Medical Practitioners, Librarians, and LIS Academics towards Health Science Library Services to Support Evidence-Based Medical Practice in South Africa.

Dear Editor,

All the authors thank you and reviewers for the constructive comments to improve the quality of the manuscript significantly. Below are the comments by the reviewers and a point-by-point response. The new paragraphs added are highlighted in green, and the revised paragraphs are highlighted in yellow. The authors have tried their best to address all the concerns and comments from the reviewers and hope the manuscript will be accepted for publication.

The title of the manuscript has been changed to "Attitudes and Opinions of Medical Practitioners, Librarians, and LIS Academics towards Health Science Library Services to Support Evidence-Based Medical Practice in South Africa," in response to the reviewers ' suggestions.

Reviewer #1: It's a good article and I don't think there's anything wrong with it, but I have two comments: the title of the article should be a little shorter, and secondly, the keywords for working with mesh should be standardized.

Response: The authors thank the reviewer for the constructive comment, which will definitely help the manuscript quality. The Title of the manuscript is revised as “Attitudes and Opinions of Medical Practitioners, Librarians, and LIS Academics towards Health Science Library Services to Support Evidence-Based Medical Practice in South Africa,” reduced from 31 words to 23 words. Line 1-2, Page 1.

The keywords are revised as Evidence-based medicine; Health science library services; Information seeking behavior; Librarians; Medical practitioners. Lines 54-55, Page 2.

Reviewer #2: This manuscript presents a comprehensive mixed-methods study exploring the support structures for Evidence-Based Medical Practice (EBMP) within health science libraries in the eThekwini district, South Africa. It addresses important gaps regarding librarian training, library resources, and practitioners’ engagement with EBMP. The topic is highly relevant to global health information science and library services, and the manuscript contributes valuable localized data with implications for both policy and LIS curriculum reform.

1. The Introduction would be strengthened by stepping further back to discuss the foundational roles of student education, university curricula, and formal training in preparing health science librarians for Evidence-Based Medical Practice (EBMP) support. Adding context about how LIS programs have (or have not) adapted to rapid technological changes, including the impacts of global pandemics and advances in electronic information resources, would help readers better understand the environment shaping current challenges. Highlighting these historical and educational dynamics early will ensure that all readers have a comprehensive perspective on the systemic factors influencing EBMP support in South Africa. Integrating recent and relevant international literature is recommended to solidify this context. It is advisable to reference relevant literature such as: 10.5334/pme.1145, https://digitalcommons.unl.edu/libphilprac/5340, 10.1080/02763869.2015.1082375, 10.1080/02763869.2024.2370756

Response: The authors thank the reviewer for the constructive suggestion. The suggested literature is added in the revised manuscript as, “Rapid technological advancements, the growth of electronic information resources, and worldwide health emergencies like the COVID-19 pandemic have also led to a significant evolution in the fundamental role that health science librarians play in supporting EBMP. For the librarian to effectively support EBMP, their formal training, university curricula, and continued professional development are crucial [14]. But there hasn't been a consistent response from Library and Information Science (LIS) programs to these developments, especially in developing countries like South Africa. Furthermore, scholarly research emphasizes the structural elements impacting health science librarians' readiness. For example, librarians must actively participate in curriculum development and assessment to support EBMP [15] and factors affecting the academic motivation of LIS students [16]. Medical librarians participate in EBMP activities pertaining to resource management and evidence dissemination, but they frequently encounter obstacles like a lack of skills, inadequate funding, and poor internet connectivity, according to research specifically from the African continent [17].” Lines 78-90, Page 3-4. The suggested literature is cited as references nos. 14-17.

And “In summary, the present study, therefore, aims to provide a comprehensive perspective on these systemic factors within the context of eThekwini, South Africa. By exploring the views of medical practitioners, librarians, and LIS academics, this research will inform a new model for health science library services that is specifically tailored to the local environment and its unique challenges. This approach acknowledges the need for LIS programs to equip future professionals with the skills necessary to navigate an ever-changing landscape.” Lines 210-216, Page 8.

2. The librarian cohort (n=5) is very small, all from public hospitals, severely limiting the representativeness and generalizability of librarian-related findings. This should be acknowledged explicitly as a study limitation and, if possible, discussed in terms of how it might bias results.

Response: This issue is addressed in section 7.7. Limited no of Library cohort: It is acknowledged the small librarian cohort (n=5), all from public hospitals, severely limiting the representativeness and generalizability of librarian-related findings. Therefore, for future studies, more librarians from public and private hospitals must be identified and included.” Lines 1123-1126, page 37.

This is discussed as, “The library cohort's small sample size (n=5) could lead to selection bias because the viewpoints and experiences of librarians employed by public hospitals might not be representative of those in private healthcare or other specialized library settings. Additionally, a small cohort might restrict the range of answers, possibly overrepresenting some points of view and underrepresenting others. The discussion of opportunities and challenges, for example, may be significantly biased toward the realities of underfunded public institutions, which may not be the case in private hospitals with substantial funding. Therefore, in order to provide a more thorough and generalizable understanding of the topic, future research should strive to include a more diverse group of librarians from both public and private institutions.” Lines 888-897, Page 30.

3. The data collection period (2014–2016) is significantly outdated considering the rapid evolution of digital health resources and library services, particularly post-COVID-19. The manuscript should better address the potential impact of this time lag on current relevance.

Response: The authors understand the reviewer concern, therefore we have included a para in the discussion section as, “Lastly, it is important to discuss whether the data collection period (2014-2016) may represent a limitation, particularly when there are major developments in digital health and library services, particularly after COVID-19. Since 2016, South Africa has developed a National Digital Health Strategy (2019-2024) aimed at creating new technology to improve healthcare services [85]. The period enforced many academic and health science libraries to familiarize with virtual services, enhance online reference services, and integrate their aids with learning management systems. Therefore, the findings of this study, while providing a valuable baseline of attitude and perceptions from the pre-pandemic era, may not fully reflect the current landscape. Thus, this study should be viewed as a foundational analysis highlighting historical challenges and opportunities that may have been aggravated or transformed by more recent events. The insights gathered, however, remain relevant for understanding the systemic factors that continue to influence EBMP support in South Africa and can inform future research into the post-pandemic state of health science libraries.” Lines 976-988, page 32-33.

4. While academic staff perspectives are included, the analysis of LIS curricula is somewhat superficial. Consider including more detailed curriculum reviews or syllabi analysis to substantiate claims regarding educational gaps.

Response: Discussed as, “A three to four year undergraduate degree or a postgraduate diploma are the two main models that LIS programs in South Africa adhere to within the framework of LIS education. The degree to which specialized subjects like EBMP are covered can be influenced by this structure, especially given how generalist many undergraduate programs are. Research methods, information literacy, and knowledge management are among the fundamental skills that are well-established in LIS curricula. Nonetheless, there seems to be a sizable lack of specialized education that particularly addresses the particular requirements of EBMP and health librarianship. The curriculum may not sufficiently address the application of these skills in clinical settings, such as using particular medical databases (e.g., PubMed, CINAHL) or navigating the complexities of institutional ethics boards and health data privacy regulations (e.g., HIPAA), even though students are taught how to conduct literature reviews and synthesize information. The results of this study are thus in line with more general discussions in the literature on LIS education, which have long argued over how to strike a balance between a generalist and a specialist curriculum. According to studies, LIS programs may still not offer the in-depth instruction needed for specialized fields like health information science, even though they have changed to incorporate more technology. Competencies in areas such as research data management, data analytics, and educational technologies, which are not always central to traditional LIS training are necessary for the changing role of the health librarian, especially in light of the growth of digital health after 2016. Lines 917-935. Page 30-31

5. The manuscript discusses differences between private and public sector settings but sometimes simplifies complex systemic factors. A more nuanced discussion could better explain disparities in resource access and practitioner attitudes.

Response: The authors thank for the constructive comment. The issue is addressed by including the text in discussion section, “Simultaneously, it is important to discuss the differences between private and public sector settings explaining disparities in resource access and practitioner attitudes. There is an unambiguous difference between the private and public sector healthcare systems in South Africa. The public sector, funded by tax revenue, serves the majority of the population (≈84%), while the private sector, funded by medical aid schemes, serves a wealthy minority (≈16%). Due to the substantial concentration of resources, such as sophisticated technology, infrastructure, and human capital, in the private sector as a result of this funding gap, the public system is consistently overburdened and underfunded. Practitioner attitudes regarding EBMP implementation may be impacted by this disparity in resources. Government healthcare workers frequently deal with serious issues like a lack of employees and obsolete equipment, which can result in high levels of stress at work, burnout, and professional frustration that could be interpreted as disengagement or a negative viewpoint [81, 82]. Conversely, practitioners in the private sector enjoy the advantages of a well-resourced setting that enables them to concentrate more on patient care and professional growth.

Private sector practitioners also benefited from having seamless access to the latest research databases, journals, and digital tools as compared to public sector practitioners who often struggle with unreliable internet connectivity and lack of access to databases, which hinders their ability to engage with new research and applying it into clinical practice [83]. Hence, these disparities in resource access and practitioner attitudes are a direct consequence of this systemic inequality. It requires fundamental changes in the health system to ensure equal distribution of resource. The proposed National Health Insurance (NHI) aims to merge the system and bridge this gap, but its execution remains a significant challenge [84]. By acknowledging these complex systemic factors, this study provides a more nuanced understanding of the challenges faced by health librarians and practitioners, highlighting the need for targeted interventions that go beyond individual professional development and address the root causes of inequality.” Line no 952-975, page 32.

6. Some sentences are overlong or complex, affecting readability. The manuscript would benefit from professional copyediting to improve clarity and conciseness.

Response: The manuscript is revised extensively by simplifying the overloaded sentences (highlighted yellow) and using Grammarly for copy editing to improve clarity and conciseness.

Reviewer #3: The present article is challenging only in terms of the statement of the problem and the title. My suggestion is to make the title shorter. This title seems too long. Secondly, I suggest that the research gap be presented in a very precise and clear manner. Overall, the rest of the article is technically and scientifically sound in my opinion.

Response: The authors thank the reviewer for the constructive comment, which will definitely help the manuscript quality. The Title of the manuscript is revised as “Attitudes and Opinions of Medical Practitioners, Librarians, and LIS Academics towards Health Science Library Services to Support Evidence-Based Medical Practice in South Africa,” reduced from 31 words to 23 words. Line 1-2, Page 1. The research gap is clearly indicated in line 46-47, page 2.

Thank you for your time and consideration. We look forward to your positive response.

Sincerely,

Dr. Saroj Bala

Department of Information and Corporate Management, Durban University of Technology

Email: sarojbalakanwal@gmail.com, Phone: +27-837856235

---

## [Editor Report · Decision Letter 1]

18 Aug 2025

Attitudes and Opinions of Medical Practitioners, Librarians, and LIS Academics towards Health Science Library Services to Support Evidence-Based Medical Practice in South Africa

PONE-D-25-34561R1

Dear Dr. Bala,

We’re pleased to inform you that your manuscript has been judged scientifically suitable for publication and will be formally accepted for publication once it meets all outstanding technical requirements.

Kind regards,

Farshid Danesh, Ph.D.

Academic Editor

PLOS ONE

---

## [Editor Report · Acceptance letter]

PONE-D-25-34561R1

PLOS ONE

Dear Dr. Bala,

I'm pleased to inform you that your manuscript has been deemed suitable for publication in PLOS ONE. Congratulations! Your manuscript is now being handed over to our production team.

Kind regards,

on behalf of

Associate Professor Farshid Danesh

Academic Editor

PLOS ONE